# Exploring Imbalanced Annotations for Effective In-Context Learning

## Abstract

Large language models (LLMs) have shown impressive performance on downstream tasks through in-context learning (ICL), which heavily relies on the quality of demonstrations selected from a large annotated dataset. However, real-world datasets often exhibit long-tailed class distributions, where a few classes occupy most of the data while most classes are under-represented. In this work, we show that imbalanced annotations hurt the ICL performance by degrading the Task Learning ability and cannot be mitigated by varying the demonstration sets, selection methods, calibration methods and rebalancing methods. To circumvent the issue, we propose a simple and effective approach termed Reweighting with Importance Factors (dubbed **RIF**) to enhance ICL performance under class imbalance. In particular, RIF constructs a balanced subset to estimate importance factors for each class: the ratio between the joint distribution of demonstration sets selected from balanced and imbalanced datasets. Then, we leverage the factors to re-weight the scoring function (e.g., the cosine similarity score used in TopK) during demonstration selection. In effect, RIF prevents over-selection from dominant classes while preserving the efficacy of current selection methods. Extensive experiments on common benchmarks demonstrate the effectiveness of our method, improving the average accuracy of current selection methods by up to 5.60%.

## 1 Introduction

Large Language Models (LLMs) have shown remarkable on downstream tasks by In-context learning (ICL) with only a few task demonstrations (Brown et al., 2020). ICL consistently outperforms zero-shot inference across various tasks without needing parameter updates, positioning it as a strong alternative to supervised fine-tuning (SFT) (Mosbach et al., 2023; Panwar et al., 2024). In particular, the success of ICL heavily relies on the quality of demonstrations selected from a large annotated dataset (Liu et al., 2022; Baldassini et al., 2024; Wang et al., 2024). However, in many real-world datasets, only a few classes (known as head classes) have an adequate number of examples, while the remaining classes (known as tail classes) are underrepresented (see Figure 4). Thus, it is of great importance to explore the effect of imbalanced annotations on in-context learning.

In this work, we study on ICL with imbalanced annotations. We empirically find that imbalanced annotations significantly hurt the ICL performance by degrading the task learning ability. Unfortunately, increasing the demonstration set or employing more effective selection methods (e.g., TopK (Liu et al., 2022) and ConE (Peng et al., 2024)) cannot mitigate the negative effect of imbalanced annotations. Existing calibration methods—such as Context Calibration (Zhao et al., 2021), which calibrates the model's output probabilities to compensate for imbalanced demonstrations-not only fail to mitigate class imbalance but can also neutralize the effectiveness of advanced selection methods. Moreover, classical rebalancing methods—such as oversampling (Chawla et al., 2002), which replicates tail-class examples until their number matches that of the head classes—yield only marginal improvements for ICL under imbalanced annotations. This motivates us to develop a universal method that consistently improves ICL's performance in the presence of imbalanced annotations.

In this paper, we show that the issue of imbalanced annotations can be resolved by reweighting the scoring function (e.g., cosine similarity for TopK (Liu et al., 2022) and cross entropy for ConE (Peng et al., 2024)) during selection. Our method, Reweighting with Importance Factors (**RIF**), is motivated by analyzing the deviation in the joint distribution of selected demonstrations induced

by imbalanced annotations. By applying importance sampling on the expected risk, we find that imbalanced annotations affect ICL's performance through the importance factor per class: the ratio between the joint distribution of demonstrations selected from imbalanced and balanced datasets.

Therefore, our key idea behind Reweighting with Importance Factors is to estimate importance factors $\boldsymbol{w}$ and incorporate them into demonstration reweighting during selection. Specifically, we construct a balanced subset by sampling an equal number of examples from each class and estimate the importance factors $\boldsymbol{w}$ via Bayesian optimization on the subset. Then, the estimated importance factors $\boldsymbol{w}$ are used to re-weight the scoring function, and the $K$ examples with the highest reweighted scores are selected as demonstrations. Empirical results show that our method prevents over-selection from dominant classes while preserving the efficacy of current selection methods.

To verify the effectiveness of our method, we conduct extensive evaluations on seven different downstream datasets, including Amazon (Keung et al., 2020), AgNews, Yelp, Yahoo (Zhang et al., 2015), Emotion (Saravia et al., 2018), NQ (Kwiatkowski et al., 2019), and CodeSearchNet (Husain et al., 2019). The results demonstrate that our method can significantly improve ICL's performance across various datasets and imbalance ratios. For example, on four classification datasets (Amazon (Keung et al., 2020), AgNews, Yelp, and Yahoo (Zhang et al., 2015)) with a 100 imbalance ratio, our method improves the average accuracy from $46.47\%$ to $52.07\%$ – a direct improvement of **5.60**%. Moreover, Section 6.1 also shows that our approach can generalize to generation tasks (e.g., NQ (Kwiatkowski et al., 2019) and CodeSearchNet (Husain et al., 2019)) to improve ICL's performance with imbalanced annotations. The code and datasets are available in the supplementary material.

Our contributions are summarized as follows:

1. We present a phenomenon: imbalanced annotations degrade ICL's performance regardless of demonstration numbers, scoring functions, calibration and rebalancing methods.

2. We propose a simple and complementary method by involving importance factors during demonstration selection to enhance ICL's performance under class imbalance. Our method is computationally efficient and agnostic to demonstration selection methods.

3. We empirically validate that our methods can improve the ICL performance in both classification and generation tasks across various imbalance ratios. Our method can be applied to both open-weight LLMs and APIs, as it only requires access to model outputs.

## 2 PRELIMINARY

### 2.1 IN-CONTEXT LEARNING

In the context of large language models (LLMs), in-context learning (ICL) aims to generate text outputs $\mathbf{y} = (y_1, ..., y_{|\mathbf{y}|})$ (i.e., token sequences) conditioned on input $\mathbf{x} = (x_1, ..., x_{|\mathbf{x}|})$ and context $\mathbf{C}_K$. In particular, the context $\mathbf{C}_K = \{(\mathbf{x}_i, \mathbf{y}_i)\}_{i=1}^K$ comprises $K$ task demonstrations (e.g. input-output pairs) selected from a large annotated dataset with $N$ examples $\mathcal{D}_s = \{(\mathbf{x}_i, \mathbf{y}_i)\}_{i=1}^N$. Let $f_\theta(\mathbf{C}_K, \cdot)$ be the ICL model with demonstrations $\mathbf{C}_K$, using the LLM $f$ parameterized by $\theta$. Given a test input $\mathbf{x}_t$, we generate the output $\mathbf{y}$ via the ICL model as:

$$\mathbf{y} = f_\theta(\mathbf{C}_K, \mathbf{x}_t) = \arg\max_{\mathbf{y}} P_\theta\left(\mathbf{y}|\mathbf{C}_K, \mathbf{x}_t\right). \tag{1}$$

To improve the performance of ICL, previous studies (Liu et al., 2022; Rubin et al., 2022; Ye et al., 2023a; Peng et al., 2024; Wang et al., 2024) designed various scoring functions $s(\cdot, \cdot)$ to select demonstrations $\mathbf{C}_K$ from an annotated dataset $\mathcal{D}_s$ as:

$$\mathbf{C}_K = \text{Top}_K\left(\{s(\mathbf{c}_i, \mathbf{x}_t)\}_{i=1}^N\right), \tag{2}$$

where $\mathbf{c}_i$ is the $i$-th example from an annotation dataset $\mathcal{D}_s$ and $\text{Top}_K(\cdot)$ denotes selecting $K$ highest-ranked examples as demonstrations from the annotation dataset $\mathcal{D}_s$ based on the given scoring function $s(\cdot, \cdot)$. For example, $\text{Top}K$ (Liu et al., 2022) selects the closest demonstrations by utilizing the cosine similarity distance between $\mathbf{x}_t$ and example $\mathbf{c}_i$.

While current selection methods showcase promising performance on commonly used benchmarks, their effectiveness may hinge on the distribution of annotated datasets $\mathcal{D}_s$. For example, ICL might

struggle to make accurate predictions for underrepresented groups within these annotated datasets. Real-world datasets (see Figure 4), however, often exhibit an imbalanced distribution, with a few 'head' classes containing many examples and numerous 'tail' classes having significantly fewer examples. The concern may lead to challenges in effectively employing in-context learning in real-world applications. We proceed with a formulation of the imbalanced setting of ICL.

## 2.2 IMBALANCED ICL

Here, we first formulate the class-imbalance setting of in-context learning in classification tasks[1], where the label space $\mathcal{Y} := \{1, \ldots, k\}$. Let $n_j$ denote the number of instances in class $j$, where $j \in \mathcal{Y}$. In the class-imbalanced setting, the annotated dataset $\mathcal{D}_s$ has an unequal distribution of instances across different classes in $\mathcal{Y}$, i.e., $n_j \ll n_k$, for some $j, k \in \mathcal{Y}$, where $j \neq k$. We quantify the imbalance ratio as $\phi = \frac{\max_{j \in \mathcal{Y}} n_j}{\min_{j \in \mathcal{Y}} n_j}$, and a higher imbalance ratio indicates a more severe class imbalance in the dataset. During test, we employ a class-balanced test dataset to ensure fair evaluation of ICL performance across classes.

In the real world, class-imbalanced distributions are frequently observed in various datasets. For instance, in the Emotion dataset (Saravia et al., 2018), the 'Joy' class constitutes 33% of the data, whereas the 'Surprise' class makes up only 4%. The CodeSearchNet dataset (Husain et al., 2019) includes 29% JavaScript while Ruby accounts for just 3%, demonstrating the significant imbalance issue. Therefore, it is crucial to ensure the performance of ICL across all classes under the class-imbalanced setting in $\mathcal{D}_s$. In the following, we analyze the effect of class imbalance in the annotated datasets from both mathematically and empirical perspectives.

## 3 PILOT STUDY

### 3.1 THE EFFECT OF IMBALANCED ANNOTATION

This section first introduces how imbalanced annotations affect the prediction of ICL from the Bayesian perspective. Let $P_c(\mathbf{x}, \mathbf{y})$ define the distribution of selected demonstrations. We begin by recalling Remark 1 (Xie et al., 2022), which states that ICL enables LLM with parameters $\theta$ to learn from the demonstration distribution $P_c(\mathbf{x}, \mathbf{y})$ through given $K$ demonstrations $\mathbf{C}_K$.

**Remark 1.** *Assume both demonstrations $\mathbf{C}_K$ and test input $\mathbf{x}_t$ are sampled from the demonstration distribution $P_c(\mathbf{x}, \mathbf{y})$. Given such demonstrations $\mathbf{C}_K$, in-context learning allows large language models $\theta$ to generate output $\mathbf{y}$ as follows:*

$$f_\theta(\mathbf{C}_K, \mathbf{x}_t) \approx \arg\max_{\mathbf{y}} P_c(\mathbf{y}|\mathbf{x}_t),$$

*where the LLMs $\theta$ generates correct output $\mathbf{y}$ following demonstration distribution $P_c(\mathbf{x}, \mathbf{y})$.*

From the Bayesian perspective, the prediction of ICL can be expressed as follows:

$$f_\theta(\mathbf{C}_K, \mathbf{x}_t) \approx \arg\max_{\mathbf{y}} P_c(\mathbf{x}_t|\mathbf{y})P_c(\mathbf{y}). \tag{3}$$

The Eq. (3) shows that the predictions of LLMs with parameter $\theta$ are biased toward the majority classes, i.e., those with larger class prior probability $P_c(\mathbf{y})$ in the demonstrations. Figure 8 illustrates a significant bias in the class prior probability $P_c(\mathbf{y})$ between demonstrations selected from imbalanced datasets and those from balanced ones, indicating that the selection bias induced by imbalanced annotations affects ICL predictions. The detailed proof is shown in Appendix D.1.

**Empirical analysis.** We also conduct experiments on various downstream tasks, including Amazon (Keung et al., 2020), AgNews, Yahoo, Yelp (Zhang et al., 2015), NQ (Kwiatkowski et al., 2019), and CodeSearchNet (Husain et al., 2019)). To simulate the class-imbalanced setting, we generate imbalanced datasets with a pre-defined probability (e.g., $\phi$ = 1, 10, 50, 100). We evaluate the performance of ICL on a balanced test dataset with various LLMs, including OPT-6.7B, -13B, -30B (Zhang et al., 2022a), LLAMA-3-8B and -70B (AI@Meta, 2024); and APIs: ChatGPT-3.5-Turbo (Achiam et al., 2023) and Gemini-2.0-Flash (Team et al., 2023). We also investigate the performance of ICL with imbalanced annotations using various sizes of demonstrations and selection methods.

---

[1]In addition to labels, the imbalance can also occur among various groups, particularly in generation tasks. We extend the imbalanced setting to generation tasks in Section 6.1.

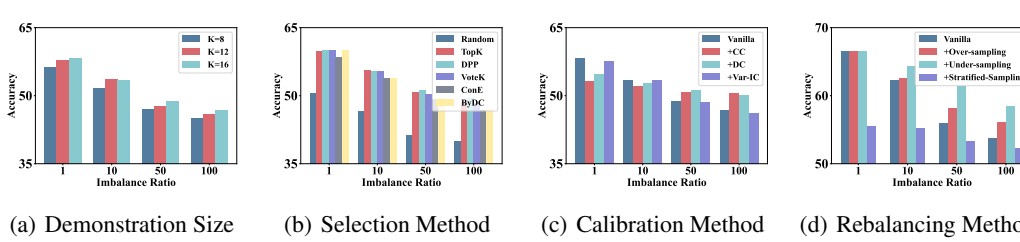

(a) Demonstration Size    (b) Selection Method    (c) Calibration Method    (d) Rebalancing Method

Figure 1: Impact of imbalanced annotations on ICL performance across four classification tasks: (a) average accuracy with varying numbers of demonstrations; (b) average accuracy with different demonstration selection methods; (c) average accuracy under different calibration methods. (d) average accuracy under different re-balancing methods.

**Imbalanced annotations significantly degrade ICL's performance.** Figure 1 (a) and (b) shows that selecting demonstrations from an imbalanced dataset significantly deteriorates ICL's performance across various imbalance ratios. Specifically, the average accuracy drops by approximately 20% for OPT-6.7B (Zhang et al., 2022a) on four different classification tasks when using six selection methods. Figure 5 shows that the decreasing trend in average accuracy is mainly due to the reduction in accuracy of the tail classes (*Business* and *Science*). Additionally, the negative effect of imbalanced annotations on ICL is observed in generation tasks, as shown in Section 6.1.

**Imbalanced annotations hurt ICL performance by degrading the Task Learning ability.** Following the previous study (Pan et al., 2023), the effect of ICL can be decomposed into two components: Task Recognition (TR) and Task Learning (TL). TR measures the extent to which LLMs can recognize a task through ICL demonstrations, whereas TL reflects the ability to capture new input–label mappings unseen in pre-training. The Figure 6 shows that TL consistently degrades as the imbalance ratio increases in ICL, while TR remains unchanged. TL relies on learning input–label mappings from the demonstrations; With the increase of imbalance rate, tail classes provide too few effective demonstrations to support such learning. In contrast, TR is insensitive to class imbalance since it mainly depends on recognizing the task format.

**The impact of demonstration number and demonstration selection.** To provide a deep understanding of imbalanced annotations, we analyze the performance of ICL across different demonstration settings, including the set size (i.e., $K$) and selection methods. Figure 1 (a) illustrates that selecting a larger set of demonstrations cannot mitigate the issue of imbalanced annotations. Meanwhile, Figure 1 (b) shows that the advantages of those powerful selection methods (e.g., TopK (Liu et al., 2022) and DPP (Ye et al., 2023b)) are neutralized in the presence of imbalanced annotations.

### 3.2 THE LIMITATION OF CALIBRATION AND REBALANCING METHODS

**Calibration methods perform poorly for ICL under imbalanced annotations.** Figure 1 (c) presents that the calibration methods yield only marginal improvements under imbalanced annotations, and may even degrade ICL performance with advanced selection methods when annotations are balanced. This is because these calibration methods merely make the model's predictive distribution revert to the distribution that a balanced demonstration set should have. However, those advanced selection methods, such as TopK (Liu et al., 2022) and DPP (Ye et al., 2023b), are more likely to select examples belonging to the same class as the test sample and conduct imbalanced demonstrations. Calibration methods inevitably degrade the performance of ICL with those advanced selection methods, thus failing to resolve the issue of imbalanced annotations.

**Re-balancing methods yield marginal improvements for ICL under imbalanced annotations.** Figure 1 (d) shows that the re-balancing methods yield only marginal improvements under imbalanced annotations, particularly at higher imbalance ratios. Specifically, oversampling uses repeated examples that may not provide additional information to LLMs for ICL performance, while under-sampling may remove key information of head classes. Stratified-Sampling undermines the effectiveness of high-performing selection methods like TopK, which tend to select demonstrations with the same class as the test input. The details of the calibration methods are provided in Appendix F.2, and those of the rebalancing methods are provided in Appendix G.1.

## 4 METHODOLOGY

The issue of imbalanced annotations originates from selection bias, where tailed classes are underrepresented in the demonstration set. Existing calibration methods cannot resolve the issue because they only consider output probabilities post-hoc, without addressing the lack of representative demonstrations for tailed classes. In this section, we first analyze the deviation in the joint distribution of demonstrations caused by imbalanced annotations. This inspires us to propose a novel method, *Reweighting with Importance Factors* (RIF), to improve the performance of ICL with imbalanced annotations through leveraging a reweighting framework during demonstration selection.

### 4.1 DEVIATION IN DEMONSTRATION JOINT DISTRIBUTION

In this section, we first present a reweighting framework for demonstration selection in ICL with imbalanced annotations. Given an annotated dataset $\mathcal{D}_s$, we define a re-weighted scoring function for demonstration selection as $s^*(\mathbf{c}_i, \mathbf{x}_t) = \frac{s(\mathbf{c}_i, \mathbf{x}_t)}{\boldsymbol{w}}$ where $s(\mathbf{c}_i, \mathbf{x}_t)$ denotes the scoring function (e.g., cosine similarity for TopK (Liu et al., 2022) and cross entropy for ConE (Peng et al., 2024)) and $\boldsymbol{w}$ denotes the class weight. Then, we choose the $K$ examples from the annotated dataset $\mathcal{D}_s$ with the highest reweighted scoring function $s^*(\mathbf{c}_i, \mathbf{x}_t)$ as demonstrations.

**Deviation of demonstration joint distribution.** Let $P_c(\mathbf{x}, \mathbf{y})$ denote the joint distribution of the demonstration set selected from the annotated dataset $\mathcal{D}_s$ using a given selection method (e.g., TopK (Liu et al., 2022) and ConE (Peng et al., 2024)) for a test dataset. We apply the importance sampling trick (Jamal et al., 2020) to connect the expected risk $\mathcal{R}_c$ over the test distribution $P_t(\mathbf{x}, \mathbf{y})$:

$$\mathcal{R}_c = \mathbb{E}_{P_t(\mathbf{x}, \mathbf{y})} M\left[f_\theta\left(\mathbf{C}_K, \mathbf{x}_t\right), \mathbf{y}_t\right] = \mathbb{E}_{P_c(\mathbf{x}, \mathbf{y})} M\left[f_\theta\left(\mathbf{C}_K, \mathbf{x}_t\right), \mathbf{y}_t\right] \frac{P_t(\mathbf{x}, \mathbf{y})}{P_c(\mathbf{x}, \mathbf{y})}$$

where $M[\cdot, \cdot]$ quantifies the discrepancy between the true output and the model's prediction. For instance, $M[\cdot, \cdot]$ denotes the error rate in classification tasks, and the negative value of the Exact Match score in question-answering tasks. Therefore, the deviation in the joint distribution of the demonstration set $P_c(\mathbf{x}, \mathbf{y})$ affects the expected risk $\mathcal{R}_c$ of in-context learning.

**Importance factors.** We define the importance factors as $\boldsymbol{w}^* = \frac{P_t(\mathbf{x}, \mathbf{y})}{P_c(\mathbf{x}, \mathbf{y})}$ where $c^*$ and $c$ denote demonstration set selected from balanced and imbalanced datasets, respectively. These factors quantify the discrepancy in the joint distribution of the demonstration set between imbalanced and balanced annotations. We establish the following equivalence of expected risks under balanced and imbalanced distributions through importance sampling:

$$\mathcal{R}_{c^*} = \mathbb{E}_{P_{c^*}(\mathbf{x}, \mathbf{y})} M\left[f_\theta\left(\mathbf{C}_K, \mathbf{x}_t\right), \mathbf{y}_t\right] \frac{P_t(\mathbf{x}, \mathbf{y})}{P_{c^*}(\mathbf{x}, \mathbf{y})} = \mathbb{E}_{P_c(\mathbf{x}, \mathbf{y})} M\left[f_\theta\left(\mathbf{C}_K, \mathbf{x}_t\right), \mathbf{y}_t\right] \frac{P_t(\mathbf{x}, \mathbf{y})}{P_c(\mathbf{x}, \mathbf{y})} \boldsymbol{w}^*,$$

which indicates that we can achieve the same expected risk of balanced annotations $R_{c^*}$ through reweighting the expected risk of imbalanced annotations $R_c$ using importance factors $\boldsymbol{w}^*$.

Section 3.1 shows that ICL achieves superior performance when demonstrations are drawn from balanced rather than imbalanced annotations. Motivated by this, we apply the importance factors $\boldsymbol{w}^*$ to reweight the scoring function during demonstration selection under imbalanced annotations, thereby reducing the expected risk:

$$\mathbb{E}_{P_c(\mathbf{x}, \mathbf{y})} M\left[f_\theta\left(\text{Top}_K\left(\left\{\frac{s(\mathbf{c}_i, \mathbf{x}_t)}{\boldsymbol{w}^*}\right\}_{i=1}^N\right), \mathbf{x}_t\right), \mathbf{y}_t\right].$$

The proof is shown in Appendix D.2.

In Figure 7, we present the average accuracy achieved by vanilla ICL, calibration methods, and reweighting with the real importance factors $\boldsymbol{w}^*$ on four classification datasets with imbalanced annotations. The results demonstrate that the reweighting approach consistently outperforms both vanilla ICL and calibration methods, highlighting its effectiveness. However, directly computing the importance factors $\boldsymbol{w}$ is non-trivial, as the joint distribution of demonstrations selected from a balanced dataset is unknown. To circumvent the issue, we aim to design an effective method to estimate the importance factors $\boldsymbol{w}$ via a balanced subset.

## 4.2 REWEIGHTING WITH IMPORTANCE FACTORS

Motivated by the previous analysis, we propose *Reweighting with Importance Factors* (RIF), a general strategy to enhance ICL performance under class imbalance. Our key idea is to estimate importance factors $w$ using a balanced subset and a re-weighting scoring function based on the estimated importance factors $w$. With this in mind, we present the details of our approach in the following.

**Selecting a balanced subset.** Given an imbalanced dataset $\mathcal{D}_s$ with $n_j$ examples for the $j$-th class, we select a balanced subset $\mathcal{D}_b$ by uniformly sampling $n_b$ examples from per class of $\mathcal{D}_s$:

$$\mathcal{D}_b = \text{UniformSample}(\mathcal{D}_s, n_b), \tag{4}$$

where $\text{UniformSample}(\mathcal{D}_s, n_b)$ denotes uniformly selecting $n_b$ examples per class of $\mathcal{D}_s$, $n_b$ is constrained by the condition $n_b < \min_{j \in \mathcal{Y}} n_j$ to ensure $n_b$ does not exceed the size of the smallest class. Through the selection, we construct a balanced subset where all classes contribute an equal number of examples, aligning with the composition of the test dataset. It is worth noting that the selected balanced subset can also be reused in the process of demonstration selection. We denote the remaining imbalanced dataset as $\mathcal{D}_r = \mathcal{D}_s - \mathcal{D}_b$, which consists of the examples remaining after removing the balanced subset $\mathcal{D}_b$ from the imbalanced dataset $\mathcal{D}_s$.

**Estimating the importance factors $w$.** As discussed above, the scaled importance factors $w$ are the one that minimizes the expected risk $\mathcal{R}_c$ under the test distribution. Then, since $\mathcal{D}_b$ shares the same joint distribution as balanced dataset $P_{c^*}(\mathbf{x}, \mathbf{y})$, we intuitively assume that the importance factors $w^*$ will also exhibit superior generalization in the balanced subset $\mathcal{D}_b$. We approximate the scaled importance factors as $w = \alpha w^* = \alpha \frac{P_c(\mathbf{x}, \mathbf{y})}{P_{c^*}(\mathbf{x}, \mathbf{y})}$ by minimizing $\mathcal{R}_c$ on the balanced subset $\mathcal{D}_b$:

$$w = \arg\min_w \frac{1}{|\mathcal{D}_b|} \sum_{i=1}^{|\mathcal{D}_b|} M\left[ f_\theta\left( \text{Top}_K\left( \left\{ \frac{s(\mathbf{c}_i, \mathbf{x}_t)}{w} \right\}_{i=1}^{|\mathcal{D}_r|} \right), \mathbf{x}_t \right), \mathbf{y}_t \right], \tag{5}$$

where we select $K$ highest-ranked demonstrations based on reweighting scoring function $s(\mathbf{c}_i, \mathbf{x}_t)$ (e.g., cosine similarity for TopK (Liu et al., 2022) and cross entropy for ConE (Peng et al., 2024)) via $w$. We estimate the importance factors $w$ using a Bayesian optimization framework (Gardner et al., 2014; Nogueira, 2014), as detailed in Appendix E, which is suitable for optimizing non-differentiable and black-box functions. In particularly, we adopt *effective numbers* (Cui et al., 2019) as initial values of the important factor $w^{(0)}$; Specifically, for class $j$, effective numbers is $w_j^{(0)} = (1 - \alpha^{n_j})/(1 - \alpha)$ where $\alpha = (N - 1)/N$ and $N$ denotes the total number of examples in $\mathcal{D}_s$.

**Re-weighting scoring function $s(\mathbf{c}_i, \mathbf{x}_t)$ with importance factors $w$.** Given a test input $\mathbf{x}_t$, we first use existing selection methods to select $K' = \lambda * K * \phi$ candidates from the imbalanced dataset $\mathcal{D}_s$ where $K$ denotes the demonstration number, $\phi$ denotes the imbalance ratio, and $\lambda$ is a control factor (e.g. $\lambda = 0.5, 1, 2$). For each candidate $\mathbf{c}_i$, we use $w$ to re-weight its scoring function $s(\mathbf{c}_i, \mathbf{x}_t)$ and employ argsort to sort the adjusted score in ascending order, producing sorted indices $\mathcal{I}$:

$$\mathcal{I} = \text{argsort}\left\{ \frac{s(\mathbf{c}_i, \mathbf{x}_t)}{w} \right\}_{i=1}^{K'}.$$

We select these candidates with $K$ low-ranking indices in sorted list $\mathcal{I}$ as the final demonstrations:

$$g(\mathbf{c}_i) = \mathbb{I}\left( \text{Loc}(\mathbf{c}_i, \mathcal{I}) \leq K \right),$$

where $\mathbb{I}$ is the indicator function and $\text{Loc}(\mathbf{c}_i, \mathcal{I})$ return the index of $\mathbf{c}_i$ in the sorted list $\mathcal{I}$. After the above steps, we establish the final $K$ demonstration set for in-context learning. Noticeably, our method offers several compelling advantages:

- **Algorithm-agnostic**: Our method can be easily incorporated into existing selection methods, improving the performance of ICL with imbalanced annotations (see Appendix G.8).

- **Easy to use**: Our method requires access only to the model outputs and integrates effortlessly with any LLMs (see Figure 2 (c)). Our method is insensitive to the size of the balance subset $|\mathcal{D}_b|$ and importance factors $w$ (see Figure 2 (a) and (b)).

- **High efficiency**: Our method requires estimating the importance factors $w$ only once for a given imbalanced dataset, without heavy hyperparameter tuning. Thus, it introduces negligible computational cost compared to other calibration methods (see Table 1).

Table 1: Average test accuracy (%) and computational cost (hours) with standard deviations (three runs) across six selection methods on four classification datasets with different imbalance ratios. The bold indicates the improvements achieved by our method. *Vanilla* refers to ICL with only existing selection methods. The detailed results for each selection method are presented in Appendix G.8.

| Dataset | Method | Imbalanced Ratios | | | | Time |
| | | 1 | 10 | 50 | 100 | |
|---------|--------|-----|-----|-----|------|------|
| AgNews | Vanilla | 80.18±0.40 | 73.96±0.63 | 64.98±1.04 | 62.95±1.29 | - |
| | CC | 76.91±0.46 | 74.24±0.83 | 72.71±1.13 | 72.38±0.83 | 4.74±0.15 |
| | DC | 78.32±0.35 | 75.35±0.81 | 71.14±0.59 | 68.46±0.88 | 4.72±0.16 |
| | Var-IC | 79.31±0.41 | 74.15±0.69 | 65.57±0.61 | 61.40±1.48 | 4.73±0.15 |
| | **Ours** | 80.16±0.46 | **79.68±0.62** | **76.29±1.09** | **75.09±1.08** | **2.83±0.10** |
| Yahoo | Vanilla | 52.96±0.48 | 50.61±0.93 | 46.91±0.88 | 44.60±1.12 | - |
| | CC | 48.50±0.51 | 48.16±1.07 | 46.37±0.83 | 46.00±0.92 | 10.61±0.29 |
| | DC | 48.74±0.90 | 47.89±0.84 | 47.77±0.83 | 46.89±0.94 | 10.71±0.23 |
| | Var-IC | 53.31±0.70 | 51.88±1.26 | 46.52±1.03 | 43.58±1.50 | 10.68±0.25 |
| | **Ours** | 53.00±0.64 | **51.72±0.90** | **49.56±1.20** | **48.86±1.53** | **6.35±0.15** |
| Amazon | Vanilla | 47.29±0.11 | 42.49±0.31 | 37.94±0.86 | 36.83±0.77 | - |
| | CC | 45.25±0.93 | 43.81±1.02 | 42.27±1.38 | 42.18±0.71 | 6.43±0.12 |
| | DC | 44.54±0.77 | 42.71±1.16 | 42.48±0.93 | 42.06±0.70 | 6.45±0.13 |
| | Var-IC | 46.88±0.47 | 43.37±0.86 | 39.29±1.34 | 38.69±0.73 | 6.52±0.08 |
| | **Ours** | 47.04±0.27 | **45.08±0.81** | **42.91±0.56** | 40.93±0.75 | **3.87±0.07** |
| Yelp | Vanilla | 48.14±0.08 | 45.81±0.55 | 42.55±0.31 | 41.51±0.34 | - |
| | CC | 41.90±0.44 | 42.47±0.56 | 42.26±0.31 | 41.61±0.36 | 6.08±0.35 |
| | DC | 47.11±0.68 | 45.16±1.29 | 43.34±0.84 | 42.83±0.87 | 6.15±0.03 |
| | Var-IC | 47.77±0.41 | 44.82±0.61 | 42.58±0.60 | 40.99±0.44 | 6.30±0.18 |
| | **Ours** | 47.91±0.16 | **46.28±0.39** | **44.07±0.56** | **43.46±0.58** | **3.76±0.22** |
| Average | Vanilla | 57.14±0.27 | 53.22±0.61 | 48.1±0.77 | 46.47±0.88 | - |
| | CC | 53.11±0.58 | 52.04±0.93 | 50.83±0.96 | 50.52±0.75 | 6.97±0.23 |
| | DC | 54.78±0.66 | 52.68±1.08 | 51.00±0.80 | 50.18±0.84 | 7.01±0.14 |
| | Var-IC | 56.82±0.50 | 53.56±0.85 | 48.43±0.92 | 46.18±1.10 | 7.06±0.17 |
| | **Ours** | 57.02±0.41 | **55.69±0.68** | **53.21±0.85** | **52.07±0.98** | **4.20±0.14** |

## 5 EXPERIMENTS

### 5.1 EXPERIMENTAL SETUP

**Models and datasets.** We utilize various large language models, including open-weight models: OPT-6.7B, OPT-13B, OPT-30B, LLAMA-3-8B, and LLAMA-3-70B; and APIs: ChatGPT-3.5-Turbo and Gemini-2.0-Flash. We use Bert-base-uncased encoder (Devlin et al., 2019) as the similarity tokenizer. For our evaluations, we verify the effectiveness of our method on four benchmark datasets, including Sentiment Classification (Amazon (Keung et al., 2020), Yelp (Zhang et al., 2015)) and Topic Classification (AgNews, Yahoo (Zhang et al., 2015)). To simulate the issue of imbalanced annotations, we generate imbalanced datasets with pre-defined imbalance ratios $\phi$ (e.g., 1, 10, 50, 100), where class frequencies follow a standard exponential distribution. Specifically, for each class $i$, the number of examples is given by $N_i = \left(\frac{N}{k}\right) \times \left(\frac{1}{\phi}\right)^{\frac{i}{k-1}}$, where $k$ is total number of classes, $N$ is total number of annotated dataset and $\phi$ denotes the imbalance ratio. We evaluate the performance of ICL on class-balanced test datasets (Shi et al., 2023; Jamal et al., 2020).

**Baselines.** We construct demonstrations for ICL using several existing selection methods, including Random (Min et al., 2022), TopK (Liu et al., 2022), DPP (Ye et al., 2023a), VoteK (Su et al., 2023), ConE (Peng et al., 2024), and ByCS (Wang et al., 2024). We also compare our method with several state-of-the-art calibration methods: CC (Zhao et al., 2021), DC (Fei et al., 2023) and Var-IC (Li et al., 2025). The details of our experiments are presented in Appendix F.

### 5.2 MAIN RESULTS

**Our method achieves superior accuracy with much less time cost.** Table 1 presents the average accuracy of ICL with different selection methods on four downstream datasets using OPT-6.7B

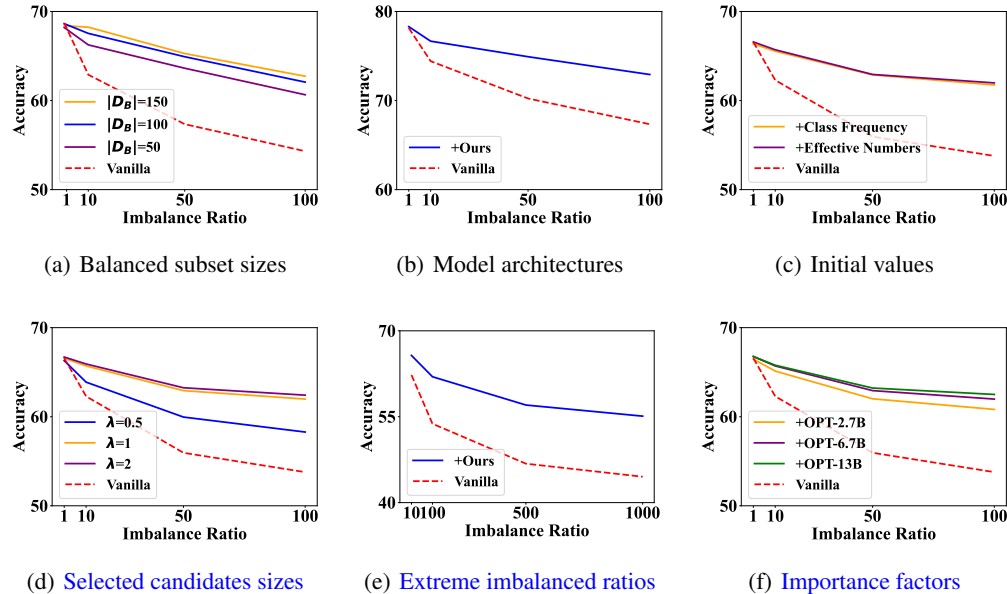

Figure 2: Results of ablation studies across six selection methods on AgNews and Yahoo with different imbalance ratios. (a) Average accuracy with different balanced subset sizes; (b) Average accuracy with different model architectures and sizes; (c) Average accuracy with different selected candidates sizes; (d) Average accuracy on datasets with extreme imbalanced ratios; (e) Average accuracy on importance factors with different initial values; (f) Average accuracy with importance factors estimated by different LLMs.

under varying imbalance ratios. A salient observation is that our method drastically improves the performance of ICL with imbalanced annotations. For example, for the 100:1 imbalance ratio, our method increases the average accuracy of ICL with six existing selection methods on AgNews from 64.02% to 75.28%, yielding an absolute improvement of **11.26** percentage points. Table 1 also reports the average computational cost across four baseline methods on different downstream datasets. Our proposed method significantly reduces the computational cost compared to existing calibration methods, while still achieving state-of-the-art downstream performance. For example, on the AgNews dataset, our approach requires only 2.83 hours compared to the suboptimal calibration method DC (4.72 hours), yielding a **50**% reduction in computational cost. The detailed results of each dataset are reported in Appendix G.8.

**Our method works with a small-scale balanced subset.** In Figure 2 (a), we examine how the size of the balanced subset $\mathcal{D}_b$ affects the effectiveness of our method (cf. Eq. 4). It's noteworthy that our method shows robustness to the choice of the size of the balanced subset $|\mathcal{D}_b|$. Even when we set $|\mathcal{D}_b| = 50$, it still yields significant improvements in ICL performance on AgNews and Yahoo datasets across six selection methods against imbalanced datasets. It is worth noting that the selected balanced subset can also be reused in the process of demonstration selection. Thus, our method is applicable in scenarios with extremely limited data from tail classes.

**Our method is effective with different model architectures and sizes.** To show our proposed method is model-agnostic, we conduct experiments on a diverse collection of model architectures and sizes, including open-weight models: OPT-6.7B, OPT-13B, OPT-30B, LLAMA-3-8B, and LLAMA-3-70B; and APIs: ChatGPT-3.5-Turbo and Gemini-2.0-Flash. The results (vanilla/+ours) in Figure 2 (b) present the same phenomenon as the main experiments in the manuscript: the ICL performance of LLMs gets worse at larger imbalanced ratios, and our method can significantly improve the performance. Thus, our method is effective on a diverse range of model architectures and sizes. The detailed results of each model are presented in the Table 7.

**Our method remains effective with different initial values.** In the paper, we employ *effective numbers* as initial value $\boldsymbol{w}^{(0)}$, which have been confirmed to successfully deal with imbalance problems in previous studies (Cui et al., 2019; Jamal et al., 2020). We employ *class frequency* $w_j = \frac{n_j}{N}$ (Shi et al., 2023; Kang et al., 2020) as an alternative initial value to verify whether initial value affects the performance of our methods. Figure 2 (c) shows that our method with *class frequency* also

improves ICL performance across six selection methods on the AgNews and Yahoo datasets under various imbalance ratios and achieves accuracy comparable to that of *effective numbers*. The results demonstrate that our method is insensitive to the choice of initial value $w^{(0)}$.

**More selected candidates $K'$ lead to better performance of our method.** We set a large value of $K'$ to ensure that all classes (especially tailed classes) are included in the subset of candidates selected by existing methods. Given a small $K'$, the candidate subset may contain too few examples from tailed classes (even missing), leading to suboptimal performance. Therefore, the $K'$ should be larger in cases of larger imbalanced ratios. We validate the claim by conducting experiments with varying $K'$ on datasets with various imbalanced ratios. The result in Figure 2 (d) demonstrates that one should set a sufficiently large $K'$ to achieve the best performance. Importantly, Table 10 shows that our method can directly reuse the scores computed by existing selection methods for the large subset of $K'$ candidates and therefore does not incur additional computational cost.

**Our method works well in the cases of extreme imbalanced ratios.** To demonstrate that our method remains effective even when tail classes contain very few examples, we conduct experiments on datasets with larger imbalanced ratios (e.g., 500, 1000), where the tail classes may have fewer than 10 examples. We first employ an LLM-based augmentation method (Li et al., 2024) to generate five times more examples that are semantically similar but phrased differently, based on the limited tail-class examples. Figure 2 (e) presents the average accuracy across six selection methods on AgNews and Yahoo datasets with extreme imbalanced ratios. The results show that our method works well in the cases of extreme imbalanced ratios (e.g. 1000). For example, when the imbalanced ratio increases from 100 to 1000, the improvement of our method increases from 8.21 to 10.58.

**The learned weights of our method exhibit cross-model transferability.** To verify the cross-model transferability of learned weights, we estimate the importance factors using various sized LLMs (OPT-2.7B, 6.7B, 13B) and evaluate them on a different model (OPT-6.7B). From the results in 2 (f) , we observed that our method can improve ICL performance using importance factors estimated by different LLMs and demonstrates strong cross-model transferability of the learned weights. For instance, reweighting selection scores using importance factors estimated by OPT-13B boosts the average test accuracy of existing selection methods from 53.78% to 62.50%, achieving an 8.72% direct improvement on the AgNews and Yahoo datasets with an imbalance ratio of 100.

# 6 DISCUSSION

## 6.1 DATA IMBALANCE IN TEXT GENERATION TASKS

Text generation, which is a common task in in-context learning, may also follow a long-tailed distribution. To address this, we verify the effect of imbalanced annotated datasets and the effectiveness of the proposed method in text generation tasks. Specifically, we consider two generation tasks: Open-Domain Question-Answering (NQ (Kwiatkowski et al., 2019)) and Code Summarization (Code-SearchNet (Husain et al., 2019)). The NQ (Kwiatkowski et al., 2019) dataset can be divided into five categories, including person (10.41%), time (20.32%), geography (9.02%), culture (45.18%), and professional knowledge (15.06%). Figure 3 (a) and (b) report the average performance of NQ and CodeSearchNet using six existing selection methods across various imbalance ratios.

Figure 3 (a) and (b) demonstrate that imbalanced annotation significantly hurts ICL's performance on generation tasks. We also observe that our method consistently improves the test performance on NQ and CodeSearchNet, indicating the generalization of our method to generation tasks. For example, our method improves the average Rouge Score of CodeSearchNet with an imbalance ratio of 100 from 28.24 to 31.30, a relative improvement of 10.83% compared to vanilla methods.

## 6.2 DOES A LARGER DATASET ADDRESS THE IMBALANCE ISSUE?

While our analysis has shown that imbalanced annotations negatively impact ICL by reducing the number of examples in tail classes, one might wonder *if this negative effect is due to the reduced size of the dataset rather than a shift in class prior distribution*. To address this, we verify the negative impact of imbalanced annotations and the effectiveness of the proposed method on such datasets with an increasing number of examples. Specifically, we consider creating an imbalanced dataset by increasing the number of examples in the head classes while keeping the number of examples in the tail classes constant. For example, in the case of Yahoo, when the imbalance ratio $\phi = 1$, the

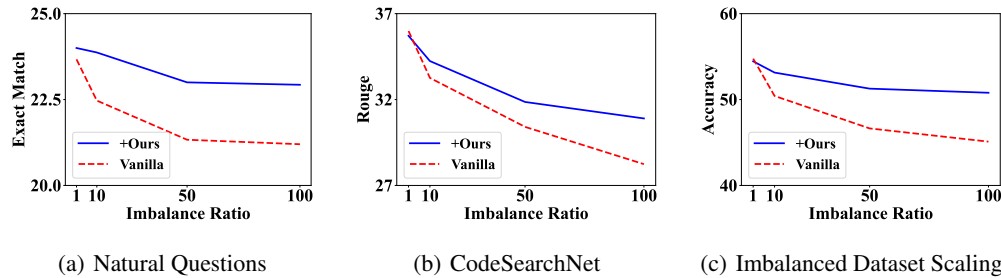

(a) Natural Questions    (b) CodeSearchNet    (c) Imbalanced Dataset Scaling

Figure 3: Figures (a) and (b) present the performance of ICL with imbalanced annotations in generation tasks (Natural Questions and CodeSearchNet), while Figure (c) shows the results on four classification tasks with increasing numbers of examples under imbalanced datasets.

number of annotated examples for each class remains 50. When the imbalance ratio $\phi = 100$, the number of examples in the head classes is 5,000, but the number in the tail classes remains 50.

Figure 3 (c) report the average accuracy of existing selection method and our method for OPT-6.7B on four classification datasets. The results show that imbalanced annotations significantly hurt the performance of ICL when more examples are added to the head classes while keeping the number of examples in the tail classes constant. This illustrates that the decreasing trend in the performance of ICL mainly depends on the class prior distribution of the annotated dataset. Notably, our method significantly improved the performance of ICL. For instance, for a 100 imbalance ratio, integrating our method outperforms vanilla ICL by a large margin of 5.43, indicating the effectiveness of our method against imbalanced annotation.

### 6.3 DOES MULTI-ATTRIBUTE IMBALANCE AFFECT IN-CONTEXT LEARNING?

In many real-world datasets, many real-world imbalances are multi-attribute or hierarchical (Beltrán et al., 2021). To explore the effect of multi-attribute annotation imbalance on ICL, we conduct experiments on datasets exhibiting imbalance across multiple attributes (topic × length). Table 2 presents the average test accuracy across six selection methods on AgNews and Yahoo datasets under two types of imbalance (topic × length). Table 2 shows that multi-attribute imbalance in annotated datasets significantly degrades the ICL performance and our method can benefit the ICL performance from a multi-attribute imbalance. For example, our method increases the average accuracy of the six selection methods on AgNews and Yahoo with topic imbalance ratio 10 and length imbalance ratio 10 from 62.29% to 65.94%.

Table 2: Average test accuracy (%) across six selection methods on AgNews and Yahoo datasets with multi-attribute imbalance ratios. The bold indicates the improved results by integrating our method. The *Vanilla* refers to the existing selection methods.

| Length Imbalanced Ratios | 1 | | 10 | |
|---|---|---|---|---|
| Topic Imbalanced Ratios | 1 | 10 | 1 | 10 |
| Vanilla/**+Ours** | 66.57/66.72 | 62.29/**65.94** | 63.25/**66.34** | 62.68/**65.22** |

## 7 CONCLUSION

In this paper, we introduce Reweighting with Importance Factors (**RIF**), a general strategy that can universally enhance the performance of in-context learning with imbalanced annotations. To the best of our knowledge, this work is the first to analyze the imbalanced annotations in ICL for both text classification and generation. Our key idea is to estimate the importance factors $w$ using a balanced subset and re-weight the scoring functions with $w$ during selection. Extensive experiments demonstrate that RIF consistently improves the performance of ICL with existing selection methods on both simulated and real long-tailed datasets. Our approach is easy to use in practice, as it is insensitive to the hyperparameters and does not introduce heavy computational cost.

**Limitations.** Our methods need to select a balanced subset from imbalanced datasets, which might be impractical if there are few examples in the tail classes. It might be an interesting direction to explore how to model the importance factors without balanced subsets.

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

## A   ETHICS STATEMENT

Our work aims to understand and mitigate the effect of imbalanced annotations on in-context learning. Regarding data and model access, all datasets and models used in this study are publicly available from prior research and do not involve any private information. This paper aims to advance research on demonstration selection for in-context learning. While our work may have various potential societal consequences, we believe none require specific emphasis here.

## B   USE OF LARGE LANGUAGE MODELS

This paper uses large language models solely to polish specific sentences, without further use of LLMs for other purposes.

## C   RELATED WORK

**In-context learning** In-context learning (ICL) is a new paradigm for large language (LLMs), which allows LLMs to make predictions only based on a few demonstrations without explicitly updating parameters (Akyurek et al., 2023; Hendel et al., 2023; Agarwal et al., 2024; Dong et al., 2024; Edwards & Camacho-Collados, 2024; Falck et al., 2024). Many studies show that ICL can achieve performance similar to fine-tuning but without the high computational cost (Gonen et al., 2023; Mosbach et al., 2023; Müller et al., 2024; Panwar et al., 2024). Despite achieving such outstanding performance, ICL has been criticized for being very sensitive to the quality of in-context examples (Fei et al., 2023; Gao et al., 2024a). Various approaches have been proposed to improve the robustness of ICL in recent years, including meta-tuning LLMs (Brunet et al., 2023), calibration (Abbas et al., 2024), demonstration selection (Zhang et al., 2022b; Nguyen & Wong, 2023; Qin et al., 2024; Ye et al., 2023b; Gao et al., 2024b; Luo et al., 2024; Mo et al., 2024), ordering (Lu et al., 2022; Liu et al., 2024), number (Zhang et al., 2025) and formation (Voronov et al., 2024; Yao et al., 2024).

The ICL is closely related to the concept of label bias (Brown et al., 2020), where language models are biased toward certain answers during few-shot learning. In particular, the most relevant part is the majority label bias, a type of context label bias that leads the model to predict answers that appear frequently in the prompt (Zhao et al., 2021; Gupta et al., 2023). An empirical work (Wang et al., 2024) suggested that label imbalance in demonstration set does not substantially impact ICL performance in binary classification. The subsequent works (Fei et al., 2023; Chu et al., 2023; Hong et al., 2024; Reif & Schwartz, 2024; Abbas et al., 2024; Li et al., 2025) delved deeper into this challenge and proposed to address it by calibrating the model's output probabilities to compensate for this bias. Additionally, other works (Hu et al., 2024; Nejjar et al., 2024) improve the fairness of ICL across classes by increasing the proportion of minority examples in the demonstration set. Some study (Chan et al., 2022) discusses how in-context learning emerges when the pre-training data exhibits particular distributional properties.While prior studies have typically focused on imbalanced distributions within prompts, our work shifts the emphasis to class imbalance in the annotation datasets from which demonstrations are drawn. Notably, this study represents the first systematic attempt to tackle the class-imbalanced issue from the perspective of demonstration selection.

**Learning with imbalanced datasets** Imbalanced datasets are common in many real-world datasets, so the challenge of class imbalance has been widely studied in the literature (Cui et al., 2018; 2019; Ochal et al., 2023; Schultheis et al., 2024). There are two popular directions in learning with imbalanced datasets: (1) Re-sampling: Re-sampling is a widely used strategy in class-imbalanced learning (Shi et al., 2023), such as Over-sampling (Chawla et al., 2002), which involves repeating data from the minority classes; Under-sampling (Liu et al., 2008), which involves removing a proportion of data from the majority classes. Stratified sampling (Vilariño et al., 2005) samples from each class have an identical probability of being sampled. In this work, we demonstrate that existing rebalancing methods yield limited improvement in in-context learning (ICL) when dealing with imbalanced annotations. (2) Training imbalance-robust models: designing loss function (Jamal et al., 2020; Tan et al., 2020; Park et al., 2023; Bhat et al., 2023; Garcin et al., 2022) or designing model architectures (Long et al., 2022; Pan et al., 2024) to mitigate the issue of imbalanced datasets. However, this method inevitably incurs high training costs that might be impractical for LLMs. In contrast, our ap-

proach only estimates the importance factors $\boldsymbol{w}$ for each class, resulting in negligible computational costs compared to traditional training methods.

# D PROOF

## D.1 PROOF 1

By Remark 1,

$$f_\theta(\mathbf{C}_K, \mathbf{x}_t) \approx \arg\max_{\mathbf{y}} P_c(\mathbf{y} \mid \mathbf{x}_t).$$

Applying Bayes' rule under the joint distribution $P_c(\mathbf{x}, \mathbf{y})$ gives

$$P_c(\mathbf{y} \mid \mathbf{x}_t) = \frac{P_c(\mathbf{x}_t \mid \mathbf{y}) \, P_c(\mathbf{y})}{P_c(\mathbf{x}_t)}.$$

Since $P_c(\mathbf{x}_t) > 0$ and does not depend on $\mathbf{y}$, the maximizer over $\mathbf{y}$ is unchanged by multiplying by the positive constant $P_c(\mathbf{x}_t)^{-1}$. Hence

$$\arg\max_{\mathbf{y}} P_c(\mathbf{y} \mid \mathbf{x}_t) = \arg\max_{\mathbf{y}} \frac{P_c(\mathbf{x}_t \mid \mathbf{y}) \, P_c(\mathbf{y})}{P_c(\mathbf{x}_t)}$$

$$\propto \arg\max_{\mathbf{y}} P_c(\mathbf{x}_t \mid \mathbf{y}) \, P_c(\mathbf{y}),$$

which is exactly Eq. 3.

## D.2 PROOF 3

For a test input $\mathbf{x}_t$ and candidate demonstration $\mathbf{c}_i$, define the original selection distribution

$$P(\mathbf{c}_i|\mathbf{x}_t) = \frac{\exp\left(\eta \cdot s(\mathbf{c}_i, \mathbf{x}_t)\right)}{\sum_{j=1}^{N} \exp\left(\eta \cdot s(\mathbf{c}_j, \mathbf{x}_t)\right)},$$

where the probabilities are normalized such that $\sum_{j=1}^{N} P(\mathbf{c}_j|\mathbf{x}_t) = 1$ and $\eta$ is a temperature parameter that controls the "sharpness" of the selection.

To minimize the expected risk $\mathcal{R}_{c^*}(P^*) = \mathbb{E}_{P_c(\mathbf{x}, \mathbf{y})} \mathbb{E}_{\mathbf{C}_K \sim P^*(\cdot|\mathbf{x})} M\left[f_\theta\left(\mathbf{C}_K, \mathbf{x}_t\right), \mathbf{y}_t\right] \frac{P_t(\mathbf{y})}{P_c(\mathbf{y})} \boldsymbol{w}$, the adjusted selection distribution $P^*(\cdot|\mathbf{x}_t)$ should: (i) favor items with low $w_i = \frac{P_c(\mathbf{x}_i, \mathbf{y}_i)}{P_{c^*}(\mathbf{x}_i, \mathbf{y}_i)}$ and (ii) preserve the effectiveness of the selection method.

To achieve goal (i) and (ii), we consider the KL-regularized objective

$$\mathcal{J} = \sum_{i=1}^{N} w_i P^*\left(\mathbf{c}_i \mid \mathbf{x}_t\right) + \lambda \operatorname{KL}\left(P^*\left(\cdot \mid \mathbf{x}_t\right) \| P\left(\cdot \mid \mathbf{x}_t\right)\right),$$

$$= \sum_{i}^{N} w_i P^*(\mathbf{c}_i|\mathbf{x}_t) + \lambda \sum_{i}^{N} P^*(\mathbf{c}_i|\mathbf{x}_t) \log \frac{P^*(\mathbf{c}_i|\mathbf{x}_t)}{P(\mathbf{c}_i|\mathbf{x}_t)},$$

subject to $\sum_{i}^{N} P^*\left(\mathbf{c}_i \mid \mathbf{x}_t\right) = 1$ and $P^*(\mathbf{c}_i \mid \mathbf{x}_t) \geq 0$. Since the expected risk $\mathcal{R}_c$ increases as the importance factors $\boldsymbol{w}$ increase, we prefer candidates with smaller weights $w_i P^*\left(\mathbf{c}_i \mid \mathbf{x}_t\right)$. $\lambda \operatorname{KL}\left(P^*\left(\cdot \mid \mathbf{x}_t\right) \| P\left(\cdot \mid \mathbf{x}_t\right)\right)$ controls the deviation from the original selection distribution.

To minimize the KL-regularized objective, we take the derivative with respect to $P^*\left(\mathbf{c}_i \mid \mathbf{x}_t\right)$ and set it equal to 0:

$$\frac{\partial \mathcal{J}}{\partial P^*\left(\mathbf{c}_i \mid \mathbf{x}_t\right)} = w_i + \lambda \left(\log \frac{P^*\left(\mathbf{c}_i \mid \mathbf{x}_t\right)}{P\left(\mathbf{c}_i \mid \mathbf{x}_t\right)} + 1\right) = 0.$$

Then, we have

$$\log P^*(\mathbf{c}_i|\mathbf{x}_t) = \log P(\mathbf{c}_i|\mathbf{x}_t) - \frac{1}{\lambda} w_i - 1.$$

Absorb constants into the normalizer

$$P^*(\mathbf{c}_i|\mathbf{x}_t) \propto P(\mathbf{c}_i|\mathbf{x}_t) \cdot \exp\left(-\frac{1}{\lambda} w_i\right).$$

Substituting $P(c_i|x_t) \propto \exp\left(\eta \cdot s(\mathbf{c}_i, \mathbf{x}_t)\right)$, we have

$$P^*(\mathbf{c}_i|\mathbf{x}_t) \propto \exp\left(\eta \cdot s(\mathbf{c}_i, \mathbf{x}_t)\right) \cdot \exp\left(-\frac{1}{\lambda} w_i\right) = \exp\left(\eta \cdot s(\mathbf{c}_i, \mathbf{x}_t) - \frac{1}{\lambda} w_i\right).$$

Consequently, selecting demonstrations based on the penalized scoring function can minimize the expected risk

$$\mathcal{R}_{c^*} = \mathbb{E}_{P_c(\mathbf{x},\mathbf{y})} M\left[f_\theta\left(\text{Top}_K\left(\left\{ s(\mathbf{c}_i, \mathbf{x}_t) - \alpha\, w_i \right\}_{i=1}^N\right), \mathbf{x}_t\right), \mathbf{y}_t\right].$$

In practice, we find that reweighting the scoring function by importance factors can also achieve similar ranking orders empirically. Thus, we consider a heuristic approximation to reduce expected risk by reweighting the scoring function by importance factors:

$$\mathbb{E}_{P_c(\mathbf{x},\mathbf{y})} M\left[f_\theta\left(\text{Top}_K\left(\left\{ \frac{s(\mathbf{c}_i, \mathbf{x}_t)}{w_i}, \right\}_{i=1}^N\right), \mathbf{x}_t\right), \mathbf{y}_t\right],$$

where $w_i = \alpha w_i^*$ with $\alpha = \frac{1}{\eta\lambda} > 0$. Notably, scaling factor $\alpha$ can be directly estimated by our method.

## E  ESTIMATING SCALED IMPORTANCE FACTORS $w$ BY BAYESIAN OPTIMIZATION

In this section, we introduce how Bayesian optimization (Gardner et al., 2014; Nogueira, 2014) can be used to estimate scaled importance factors $\boldsymbol{w}$:

$$\boldsymbol{w} = \arg\min_{\boldsymbol{w}} \frac{1}{|\mathcal{D}_b|} \sum_{i=1}^{|\mathcal{D}_b|} M\left[f_\theta\left(\text{Top}_K\left(\left\{ \frac{s(\mathbf{c}_i, \mathbf{x}_t)}{\boldsymbol{w}} \right\}_{i=1}^{|\mathcal{D}_r|}\right), \mathbf{x}_t\right), \mathbf{y}_t\right]. \quad (6)$$

**Surrogate Model.** We use the Gaussian Process as a surrogate model to approximate the objective function 6. Initially, a prior distribution is established in a general form for the estimated model parameters,

$$\mathcal{F}_\theta(\boldsymbol{w}) = \mathcal{N}\left(\mu(\boldsymbol{w}), \sigma^2(\boldsymbol{w})\right), \quad (7)$$

where $\mathcal{N}$ is the Gaussian distribution with a mean function $\mu(\boldsymbol{w})$ and a covariance function $\sigma^2(\boldsymbol{w})$. For a data point $\boldsymbol{w}_j$ sampled from the Gaussian process $\mathcal{N}$, we compute its corresponding function values $\mathcal{F}_\theta(\boldsymbol{w}_j)$.

**Acquisition Function.** We use the Expected Improvement (EI) criterion to select the point $\boldsymbol{w}_{j+1}$ with the maximum expected improvement as the next query point:

$$\text{EI}(\boldsymbol{w}) = \begin{cases} \left(\mu_j(\boldsymbol{w}) - \mathcal{F}_\theta(\boldsymbol{w}^+) - \epsilon\right) \Phi(Z) + \sigma_j(\boldsymbol{w})\phi(Z) & \text{if } \sigma_j(\boldsymbol{w}) > 0, \\ 0 & \text{if } \sigma_j(\boldsymbol{w}) = 0. \end{cases} \quad (8)$$

$$Z = \frac{\mu_j(\boldsymbol{w}) - \mathcal{F}_\theta(\boldsymbol{w}^+) - \epsilon}{\sigma_j(\boldsymbol{w})}, \quad (9)$$

where $\mu_j(\boldsymbol{w})$ and $\sigma_j(\boldsymbol{w})$ represent the mean and standard deviation of the surrogate model at step $j$, $\boldsymbol{w}^+$ denotes the current optimal observed point, $\epsilon$ is a tunable hyper-parameter. $\Phi(Z)$ and $\phi(Z)$ are the probability density function and cumulative density function of the Gaussian Process.

**Iterative Optimization Process.** We start with an initial set of observations $\boldsymbol{w}^{(0)}$ by evaluating the objective function $\{\mathcal{F}(\boldsymbol{w}_i)\}_{i=0}^h$ at a few selected points. Second, we use the observed data $(\boldsymbol{w}_j, \mathcal{F}(\boldsymbol{w}_j))$ to update the Gaussian Process model, refining the estimates of $\mu_j(\boldsymbol{w})$ and $\sigma_j(\boldsymbol{w})$. Third, we employ the *Expected Improvement* criterion to determine the next point $\boldsymbol{w}_{j+1}$ and obtain $\{\mathcal{F}(\boldsymbol{w}_{j+1})\}_{i=0}^h$. We continue the process of updating the model and selecting new points until a stopping criterion is met. Finally, we select the point $\boldsymbol{w}^+$ with the best observed value of objective function $\mathcal{F}_\theta(\boldsymbol{w}^+)$ as the optimal solution. We set the maximum number of iterations to 30.

# F  EXPERIMENTAL SETTING

## F.1  DEMONSTRATION SELECTION METHODS

To verify the effectiveness of our method, we consider both learning-free and other learning-based retrievers as baselines, including **Random** selects demonstrations randomly from an annotated dataset without repetition (Min et al., 2022); **TopK** retrieves demonstrations that are semantically similar to test inputs (Liu et al., 2022); **DPP** employ Determinantal Point Processes (DPPs) to model the interaction between the test input and demonstrations (Ye et al., 2023a); **VoteK** proposes an unsupervised and graph-based selective annotation method to select diverse and representative demonstrations (Su et al., 2023); **ConE** searches for demonstrations by minimizing the difference in cross-entropy between the test input and the demonstrations (Peng et al., 2024); **ByCS** assumes that an accurate inverse likelihood probability will lead to an accurate posterior probability and selects demonstrations based on their inverse inference results (Wang et al., 2024).

## F.2  CALIBRATION METHODS

Some calibration methods (Zhao et al., 2021; Fei et al., 2023; Li et al., 2025) have been proposed to mitigate the LLMs' inherent biases in ICL prediction, where LLMs exhibit a preference for certain answers induced by imbalanced demonstration. Imbalanced annotation denotes a skewed joint distribution across an annotated dataset, whereas imbalanced demonstration refers to the skew within a small set of samples selected from an annotated dataset. In this section, we examine whether three calibration methods—Contextual Calibration (Zhao et al., 2021), Domain-context Calibration (Fei et al., 2023) and Variation of In-Context Examples (Li et al., 2025) — can mitigate the issue of imbalanced annotations in ICL. Specifically, we estimate and correct label bias by follows:

$$\mathbf{y} = \arg\max_{\mathbf{y}} \frac{P_\theta\left(\mathbf{y}|\mathbf{C}_K, \mathbf{x}_t\right)}{\bar{P}_\theta\left(\mathbf{y}|\mathbf{C}_K, \mathbf{x}_{cf}\right)},$$

where $P_\theta\left(\mathbf{y} \mid \mathbf{C}_K, \mathbf{x}_t\right)$ denotes the probability assigned by the LLM with parameters $\theta$ to label $\mathbf{y}$ given the test input $\mathbf{x}_t$ and demonstrations $\mathbf{C}_K$, while $\bar{P}_\theta\left(\mathbf{y} \mid \mathbf{C}_K, \mathbf{x}_{cf}\right)$ denotes the average probability of label $\mathbf{y}$ conditioned on the content-free words or sentences $\mathbf{x}_{cf}$ with the same demonstrations $\mathbf{C}_K$. **Contextual Calibration** (CC) estimates a label bias using content-free inputs (e.g., 'N / A') as $\mathbf{x}_{cf}$ and rescales $P_\theta\left(\mathbf{y} \mid \mathbf{C}_K, \mathbf{x}_t\right)$ (Zhao et al., 2021). **Domain-context Calibration** (DC) samples random in-domain words from a bag-of-words to form content-free inputs (Fei et al., 2023). **Variation of In-Context Examples** (Var-IC) varies the selection and ordering of demonstrations, treating each variation as an ensemble component to produce calibrated predictions (Li et al., 2025).

## F.3  INFERENCE

For classification tasks, we compute the sentence perplexity for each sequence formed by concatenating the input with each candidate answer (Brown et al., 2020; Wu et al., 2023). Specifically, for each input instance $\mathbf{x}$ is paired potential label set $\mathcal{Y}$, where $\mathcal{Y}$ represents the set of possible classes (e.g., Sports, Business, etc.). Then, for each possible label $\mathbf{y} \in \mathcal{Y}$, we concatenate each tokenized input-output pair $(\mathbf{x}, \mathbf{y})$, and obtain the corresponding tokenized sequence $\mathbf{c} = (z_1, ..., z_{|\mathbf{c}|}) = (x_1, ..., x_{|\mathbf{x}|}, y_1, ..., y_{|\mathbf{y}|})$, where $|\mathbf{c}| = |\mathbf{x}| + |\mathbf{y}|$. Now, the perplexity of $\mathbf{c}$ is calculated as:

$$\text{Perplexity}(\mathbf{c}) = \exp\{-\frac{1}{|\mathbf{c}|} \sum_{i=1}^{|\mathbf{c}|} \log P_\theta(c_i|c_{<i})\},$$

where $\log P_\theta(c_i|c_{<i})$ is the log-likelihood of the $i$-th token conditioned on the preceding tokens $c_{<i}$, from the given language model parameterized by $\theta$. We select the label corresponding to the input-output pair $\mathbf{c}$ with the lowest perplexity as the predicted label for the input $\mathbf{x}$.

For text generation tasks, we represent candidate answers using tokens from the vocabulary and select the final prediction based on the one with the highest probability (Brown et al., 2020; Wu et al., 2023). To reduce computational cost, we set the maximum new tokens to 50 and limit unnecessary token generation.

### F.4    DATASETS

We conduct experiments on various classification and generation tasks and examples of each dataset are shown in Tables 15. We collect all datasets from Huggingface. The train sets are regarded as annotated dataset and the test datasets are used to evaluate the performance of ICL. We generate categories for Natural question (Kwiatkowski et al., 2019) using ChatGPT-3.5-Turbo. The categories include people, time, geography, culture, and specialized knowledge.

### F.5    EXPERIMENT DETAILS

We run our experiments on NVIDIA GeForce RTX 4090 and NVIDIA L40 GPU, and implement all methods by *PyTorch* and *transformers*. Our code is inspired by OpenICL Wu et al. (2023). We thank the authors for releasing their code.

## G    MORE EMPIRICAL RESULTS

### G.1    CAN OUR METHOD OUTPERFORM OTHER RE-BALANCING METHODS?

One simple and intuitive approach to deal with the class-imbalanced problem is rebalancing (Shi et al., 2023). In this section, we examine whether three classical rebalancing methods—over-sampling (Chawla et al., 2002), under-sampling (Liu et al., 2008), stratified sampling (Vilariño et al., 2005) and reweighting (Cui et al., 2018)—can mitigate the negative effects of imbalanced annotations on ICL. Specifically, we select top $K$ examples ranked by the score function $s(\cdot, \cdot)$ from an annotated dataset with $N$ examples as demonstrations:

$$\mathbf{C}_K = \text{Top}_K \left( \left\{ s(\mathbf{c}_i^{'}, \mathbf{x}_t) \right\}_{i=1}^{N^{'}} \right), \tag{10}$$

$$\mathbf{C}_K = \sum_{i=1}^{k} \text{top}_{\frac{K}{k}} \left( \{ s(\mathbf{c}_j, \mathbf{x}_t) \}_{j=1}^{n_i} \right). \tag{11}$$

$$\mathbf{C}_K = \text{Top}_K \left( \left\{ \frac{s(\mathbf{c}_i, \mathbf{x}_t)}{w_i} \right\}_{i=1}^{N} \right). \tag{12}$$

For the over-sampling method in Eq. (10), we select demonstrations $\mathbf{c}_i^{'}$ from an over-sampling dataset with $N^{'}$ examples, where we repeat the examples of tail classes until their number matches that of head classes. The under-sampling method in Eq. (10) randomly trims examples belonging to head classes until the number of examples in head classes is equal to that of tail classes. Suppose $n_j$ represents the size of the class to which the $i$-th example belongs, stratified sampling in Eq. (11) selects $\frac{K}{k}$ demonstrations from each class with $n_j$ examples. For the re-weighting method in Eq. (12), we select the top $K$ examples based on the scoring functions $s(\mathbf{c}_i, \mathbf{x}_t)$ multiplied by the class weights without our method.

Table 3: Average test accuracy (%) of AgNews and Yahoo across six selection methods with rebalancing and our methods. The bold indicates the improved results by integrating our method. The *Vanilla* refers to the existing selection methods.

| Method | Imbalanced Ratios | | | |
|---|---|---|---|---|
| | 1 | 10 | 50 | 100 |
| Vanilla | 66.57 | 62.29 | 55.95 | 53.78 |
| +Over-Sampling | 66.57 | 62.56 | 58.08 | 56.08 |
| +Under-Sampling | 66.57 | 64.28 | 61.34 | 59.36 |
| +Stratified-Sampling | 55.42 | 55.25 | 53.29 | 52.21 |
| +Re-weighting (Normalized Class Freque | 66.57 | 62.68 | 57.45 | 55.68 |
| +Re-weighting (Effective Number) | 66.57 | 62.98 | 57.60 | 55.84 |
| **+Ours** | **66.72** | **65.94** | **63.14** | **62.22** |

Table 3 shows that the rebalancing methods can only achieve limited effects on AgNews and Yahoo datasets. Intuitively, oversampling uses repeated examples that may not provide additional information to LLMs for ICL performance, while undersampling may remove key information of head classes. For instance, with an imbalanced ratio of 100, over-sampling boosts the performance of ICL from 62.95 to 65.50, yielding limited improvement. In contrast, stratified sampling, which selects demonstrations equally from each class, reduces the performance of ICL from 62.95 to 68.80, as it undermines the effectiveness of high-performing selection methods like TopK (Liu et al., 2022).

## G.2 RESULTS WITH BOARDER BASELINES.

Here, we compare with more baselines, including FCG (Hu et al., 2024), over-sampling+CC, under-sampling+CC and stratified-sampling+CC. Table 4 presents average test accuracy of the baselines and our method on the two classification tasks: AgNews and Yahoo. The results show that our method can outperform these ensemble baselines.

Table 4: Average test accuracy (%) across six selection methods on AgNews and Yahoo datasets with different iterations. The bold indicates the improved results by integrating our method. The *Vanilla* refers to the existing selection methods.

| Method | Imbalanced Ratios | | | |
|---|---|---|---|---|
| | 1 | 10 | 50 | 100 |
| Vanilla | 66.57 | 62.29 | 55.95 | 53.78 |
| FCG | 63.34 | 62.64 | 61.25 | 60.50 |
| +Over-sampling+CC | 62.71 | 60.44 | 59.02 | 58.65 |
| +Under-sampling+CC | 62.71 | 60.72 | 59.68 | 59.28 |
| +Stratified-sampling+CC | 61.30 | 61.20 | 60.88 | 60.27 |
| +Ours | 66.72 | 65.94 | 63.14 | 62.22 |

## G.3 IS OUR METHOD ROBUST WITH DIFFERENT ITERATIONS?

The Bayesian optimization employed in our method is well-suited for efficiently optimizing non-differentiable and black-box functions with a few iterations (Gardner et al., 2014; Nogueira, 2014). Here, we conduct experiments on our method with different iterations (e.g., 10, 30 and 50). The table 5 presents the average test accuracy across six selection methods on AgNews and Yahoo datasets with different imbalance ratios. The results show that benefit the ICL performance from a few iteration (e.g. 10). In fact, our method can achieve significant improvement with minimal computational cost using 30 iterations.

Table 5: Average test accuracy (%) across six selection methods on AgNews and Yahoo datasets with different iterations. The bold indicates the improved results by integrating our method. The *Vanilla* refers to the existing selection methods.

| Method | Imbalanced Ratios | | | |
|---|---|---|---|---|
| | 1 | 10 | 50 | 100 |
| Vanilla | 66.57 | 62.29 | 55.95 | 53.78 |
| +Ours (Iteration=10) | 66.24 | 64.32 | 60.62 | 59.48 |
| +Ours (Iteration=30) | 66.58 | 65.70 | 62.93 | 61.98 |
| +Ours (Iteration=50) | 66.72 | 65.94 | 63.14 | 62.22 |

## G.4 RESULTS OF DIFFERENT PROMPT FORMATS AND VERBALIZER CHOICES

Here, we conduct exmperiments with different prompt formats and verbalizer choices. The results (vanilla/ours) in the table 6 present the same pheonomenon as main experiments in the manuscript: the ICL performance of LLMs get worse at larger imbalanced ratios, and our method can significantly improve the performance, especially at large imbalanced ratios (e.g., 100). We add the sensitivity analysis of hyperparameters in Appendix D.3 of the revised version.

## G.5 PERFORMANCE ON DIFFERENT MODEL ARCHITECTURES AND SIZES

We conduct experiments on various sized LLMs (including open-weight models and APIs). The results (vanilla/ours) in the table 7 below present the same phenomenon as the main experiments

Table 6: Average test accuracy (%) across six selection methods on AgNews and Yahoo datasets with different with prompt formats and verbalizer choices. The bold indicates the improved results by integrating our method. The *Vanilla* refers to the existing selection methods.

| Method | Imbalanced Ratios | | | |
|---|---|---|---|---|
| | 1 | 10 | 50 | 100 |
| Format 1 | 66.57/66.58 | 62.29/65.70 | 55.95/62.93 | 53.78/61.98 |
| Format 2 | 67.17/67.13 | 63.65/65.52 | 58.21/63.94 | 56.33/63.33 |
| Format 3 | 66.97/66.70 | 62.05/65.24 | 55.51/62.52 | 53.13/61.33 |

in the manuscript: the ICL performance of LLMs get worse at larger imbalanced ratios, and our method can significantly improve the performance, especially at large imbalanced ratios (e.g., 100).

Table 7: Average accuracy (%) of Vanilla / **+Ours** methods on the AgNews and Yahoo datasets with different model architectures and sizes. Bold numbers are superior results.

| Imbalance Ratio | 1 | 10 | 50 | 100 |
|---|---|---|---|---|
| | | Vanilla/**+Ours** | | |
| OPT-6.7B | 68.71/68.63 | 62.90/**67.54** | 57.35/**64.92** | 54.31/**62.07** |
| OPT-13B | 72.22/72.84 | 68.04/**70.12** | 62.28/**68.16** | 58.12/**64.33** |
| OPT-30B | 76.34/76.60 | 72.32/**74.10** | 67.74/**71.90** | 64.30/**70.08** |
| LLAMA-3-8B | 78.90/79.10 | 75.20/**77.16** | 69.32/**75.82** | 65.72/**73.20** |
| LLAMA-3-70B | 86.50/86.43 | 83.07/**84.80** | 79.86/**83.13** | 77.60/**81.22** |
| ChatGPT-3.5-Turbo | 82.45/82.40 | 79.47/**81.08** | 77.88/**80.20** | 76.32/**79.74** |
| Gemini-2.0-Flash | 81.91/82.09 | 79.95/**81.86** | 77.19/**80.31** | 75.01/**79.82** |

G.6 DISCUSSION OF IMBALANCED TEST SET

We also conducted experiments on an imbalanced test set. The table below presents the average Macro-F1 metric on AgNews. The results in Table 8 demonstrate that our method can improve ICL performance on an imbalanced test set. For example, for 100 imbalance ratio, our approach improves the test Macro-F1 of vanilla ICL from 55.70 to 59.77 – a 4.07 of direct improvement.

Table 8: Average Macro-F1 metric of AgNews across six selection methods of Vanilla ICL, calibration and our methods. The bold indicates the improved results by integrating our method. The *Vanilla* refers to the existing selection methods.

| Dataset | Method | Imbalanced Ratios | | | |
|---|---|---|---|---|---|
| | | 1 | 10 | 50 | 100 |
| AgNews | Vanilla | 80.83 | 71.30 | 62.63 | 55.70 |
| | +CC | 77.56 | 70.30 | 56.82 | 48.26 |
| | +DC | 78.99 | 73.47 | 59.72 | 52.02 |
| | +Var-IC | 82.89 | 74.25 | 64.79 | 57.44 |
| | **+Ours** | 80.89 | **76.52** | **66.71** | **59.77** |

G.7 CAN OUR METHOD IMPROVE REAL-WORLD IMBALANCED DATASET

We verify the effectiveness of our method on a real-world imbalanced dataset. The Emotion (Saravia et al., 2018) dataset has a long-tailed distribution (refer to Figure 4). Table 9 shows that our method consistently improves existing selection methods and outperforms existing calibration methods, including Contextual Calibration (Zhao et al., 2021), Domain-context Calibration (Fei et al., 2023) and Variation of In-Context Examples (Li et al., 2025). For example, with OPT-6.7B (Zhang et al., 2022a), using our method boosts the average Macro-F1 of six selection methods from 32.12 to 36.69, a direct improvement of 4.57. These results verify that our method is effective in improving ICL's performance in real-world imbalanced scenarios.

G.8 MAIN RESULTS FOR EACH DATASET

Tables 11, 12, 13 and 14 report average test accuracy (%) with standard deviation on AgNews, Yahoo, Amazon and Yelp datasets with various imbalanced ratios (over 3 runs), respectively.

Table 9: Average Macro-F1 metric on Emotion dataset. Bold numbers are superior results.

| Methods | Vanilla | +CC | +DC | +Var-IC | +Ours |
|---|---|---|---|---|---|
| Accuracy | 32.22 | 34.02 | 34.72 | 35.56 | 36.69 |

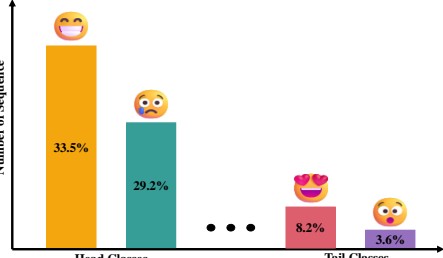

Figure 4: **An example of imbalanced dataset.** In Emotion (Saravia et al., 2018), a few sentiments make a large contribution and data tend to show a long-tailed distribution. For example, "joy" and "sadness" are head classes, while most other classes, such as "love" and "surprise", are tail classes.

Table 10: Average computational cost of AgNews and Yahoo across six selection methods and our methods.

| Method | Imbalanced Ratios | | | |
|---|---|---|---|---|
| | 1 | 10 | 50 | 100 |
| Vanilla | 0.19 | 0.30 | 1.46 | 3.01 |
| $\lambda = 0.5$ | 0.10 | 0.15 | 0.74 | 1.50 |
| $\lambda = 1$ | 0.20 | 0.31 | 1.48 | 3.05 |
| $\lambda = 2$ | 0.40 | 0.64 | 3.06 | 6.07 |

Table 11: Average test accuracy (%) and computational cost (hours) with standard deviations (three runs) on AgNews under different imbalance ratios. The bold indicates the improved results by integrating our method. The *Vanilla* refers to the existing selection methods.

| Dataset | Method | Imbalanced Ratios | | | | Time |
| | | 1 | 10 | 50 | 100 | |
|---|---|---|---|---|---|---|
| AgNews | Random | 66.20±0.40 | 61.27±0.89 | 50.17±1.42 | 49.63±1.22 | - |
| | +CC | 71.47±0.51 | 68.37±0.44 | 65.70±0.80 | 65.67±0.56 | 4.89±0.12 |
| | +DC | 72.87±0.44 | 67.90±0.80 | 65.33±1.16 | 64.80±1.33 | 4.85±0.12 |
| | +Var-IC | 67.73±0.51 | 61.30±0.93 | 48.77±1.02 | 45.57±1.56 | 4.86±0.23 |
| | **+Ours** | 66.43±0.91 | **67.43±0.24** | **67.83±0.29** | **67.50±1.27** | **2.96±0.03** |
| | TopK | 83.80±0.00 | 75.77±0.31 | 68.97±0.42 | 65.40±1.67 | - |
| | +CC | 77.87±0.44 | 74.80±1.20 | 73.77±1.42 | 73.63±1.11 | 4.65±0.10 |
| | +DC | 79.90±0.27 | 76.37±1.09 | 71.43±0.51 | 69.77±0.76 | 4.59±0.12 |
| | +Var-IC | 81.63±0.18 | 76.40±0.00 | 68.63±0.64 | 64.63±1.78 | 4.64±0.18 |
| | **+Ours** | 83.50±0.40 | **82.57±0.31** | **78.20±1.00** | **76.3±1.07** | **2.82±0.09** |
| | DPP | 83.20±0.40 | 76.07±0.16 | 68.3±0.87 | 65.33±1.11 | - |
| | +CC | 78.47±0.49 | 75.13±1.42 | 74.03±1.24 | 74.30±0.67 | 4.71±0.22 |
| | +DC | 79.57±0.36 | 76.37±1.09 | 72.1±0.13 | 69.43±0.98 | 4.66±0.22 |
| | +Var-IC | 81.47±0.31 | 77.07±0.89 | 68.30±0.27 | 64.30±1.33 | 4.56±0.07 |
| | **+Ours** | 83.23±0.38 | **82.63±1.09** | **77.90±0.93** | **76.57±0.89** | **2.75±0.11** |
| | VoteK | 82.67±0.44 | 76.73±0.84 | 67.4±1.47 | 65.87±1.31 | - |
| | +CC | 77.80±0.40 | 75.47±0.98 | 74.70±0.80 | 74.67±0.42 | 4.60±0.17 |
| | +DC | 79.23±0.18 | 77.03±0.64 | 73.10±0.73 | 69.70±0.80 | 4.87±0.14 |
| | +Var-IC | 81.83±0.56 | 77.07±0.89 | 69.10±0.53 | 65.63±1.56 | 4.76±0.14 |
| | **+Ours** | 82.50±0.33 | **81.97±0.64** | **78.13±1.44** | **76.93±0.76** | **2.75±0.15** |
| | ConE | 82.30±0.60 | 77.17±0.69 | 67.2±1.33 | 65.50±1.73 | - |
| | +CC | 77.47±0.49 | 75.83±0.56 | 74.03±1.24 | 73.67±0.84 | 4.78±0.12 |
| | +DC | 78.77±0.44 | 77.40±0.27 | 72.43±0.51 | 69.03±0.18 | 4.71±0.02 |
| | +Var-IC | 81.43±0.42 | 76.80±0.53 | 69.30±0.60 | 64.63±1.11 | 4.82±0.16 |
| | **+Ours** | 82.60±0.40 | **81.77±0.76** | **77.5±1.67** | **76.80±1.20** | **2.79±0.11** |
| | ByCS | 82.90±0.53 | 76.77±0.91 | 67.87±0.76 | 65.97±0.71 | - |
| | +CC | 78.40±0.40 | 75.83±0.36 | 74.03±1.24 | 72.33±1.36 | 4.79±0.16 |
| | +DC | 79.60±0.40 | 77.03±0.96 | 72.43±0.51 | 68.03±1.24 | 4.61±0.32 |
| | +Var-IC | 82.10±0.47 | 76.27±0.89 | 69.3±0.6 | 63.63±1.56 | 4.75±0.11 |
| | **+Ours** | 82.67±0.36 | **81.73±0.69** | **78.17±1.22** | **76.47±1.29** | **2.81±0.14** |
| Average | Vanilla | 80.18±0.40 | 73.96±0.63 | 64.98±1.04 | 62.95±1.29 | - |
| | +CC | 76.91±0.46 | 74.24±0.83 | 72.71±1.13 | 72.38±0.83 | 4.74±0.15 |
| | +DC | 78.32±0.35 | 75.35±0.81 | 71.14±0.59 | 68.46±0.88 | 4.72±0.16 |
| | +Var-IC | 79.31±0.41 | 74.15±0.69 | 65.57±0.61 | 61.40±1.48 | 4.73±0.15 |
| | **+Ours** | 80.16±0.46 | **79.68±0.62** | **76.29±1.09** | **75.09±1.08** | **2.83±0.10** |

Table 12: Average test accuracy (%) and computational cost (hours) with standard deviations (three runs) on Yahoo under different imbalance ratios. The bold indicates the improved results by integrating our method. The *Vanilla* refers to the existing selection methods.

| Dataset | Method | Imbalanced Ratios | | | | Time |
| | | 1 | 10 | 50 | 100 | |
|---|---|---|---|---|---|---|
| Yahoo | Random | 44.20±0.59 | 44.47±0.81 | 39.27±1.18 | 37.60±0.59 | - |
| | +CC | 51.53±0.98 | 52.27±0.62 | 51.80±1.33 | 51.67±1.56 | 10.75±0.34 |
| | +DC | 52.93±1.78 | 51.40±0.67 | 50.97±2.09 | 48.90±2.20 | 10.42±0.18 |
| | +Var-IC | 47.87±0.44 | 44.63±1.44 | 40.20±1.33 | 37.63±1.04 | 10.53±0.31 |
| | **+Ours** | 43.20±1.18 | 42.93±0.25 | 44.33±0.74 | 44.00±1.93 | **6.50±0.15** |
| | TopK | 53.67±0.19 | 52.93±1.27 | 50.27±0.90 | 47.27±1.32 | - |
| | +CC | 47.23±0.38 | 46.80±1.73 | 45.20±0.67 | 44.07±0.58 | 10.86±0.1 |
| | +DC | 46.97±0.78 | 46.20±2.00 | 47.27±0.36 | 46.47±1.11 | 10.69±0.44 |
| | +Var-IC | 55.53±1.02 | 54.80±2.13 | 46.97±0.82 | 45.40±1.20 | 10.86±0.21 |
| | **+Ours** | 54.40±0.85 | **56.00±1.50** | **50.53±1.33** | **50.20±2.70** | **6.43±0.16** |
| | DPP | 56.57±0.26 | 54.03±0.97 | 50.47±0.57 | 48.87±1.25 | - |
| | +CC | 48.60±0.27 | 48.00±1.20 | 44.90±0.47 | 45.73±1.02 | 10.38±0.21 |
| | +DC | 48.83±0.82 | 47.53±0.89 | 47.13±0.76 | 46.73±0.62 | 10.65±0.39 |
| | +Var-IC | 54.87±0.62 | 54.13±1.24 | 47.37±1.22 | 45.13±1.58 | 10.48±0.22 |
| | **+Ours** | 56.07±0.19 | **54.87±0.68** | **51.07±1.46** | **51.60±1.82** | **6.28±0.14** |
| | VoteK | 56.37±0.45 | 53.10±0.54 | 48.70±1.08 | 46.20±0.82 | - |
| | +CC | 48.50±0.33 | 47.67±1.04 | 45.90±0.40 | 44.40±1.47 | 10.37±0.18 |
| | +DC | 48.57±0.91 | 48.07±0.38 | 47.47±0.31 | 46.40±0.53 | 10.65±0.38 |
| | +Var-IC | 54.63±0.22 | 53.80±0.80 | 48.03±0.89 | 45.13±1.58 | 10.48±0.10 |
| | **+Ours** | 56.03±0.26 | **55.10±1.10** | **52.03±1.31** | **51.07±0.94** | **6.27±0.11** |
| | ConE | 51.90±0.80 | 49.37±0.90 | 45.33±0.68 | 42.77±1.47 | - |
| | +CC | 46.83±0.91 | 46.67±0.89 | 44.37±1.44 | 44.73±0.36 | 10.74±0.3 |
| | +DC | 46.90±0.60 | 46.73±0.51 | 46.60±1.07 | 46.07±0.58 | 10.90±0.18 |
| | +Var-IC | 53.30±1.00 | 51.13±1.38 | 46.83±0.91 | 42.80±1.53 | 10.83±0.18 |
| | **+Ours** | 52.90±0.67 | **49.37±0.90** | **49.00±1.50** | **47.43±0.83** | **6.29±0.07** |
| | ByCS | 55.03±0.58 | 49.77±1.11 | 47.40±0.86 | 44.87±1.25 | - |
| | +CC | 48.30±0.20 | 47.53±0.96 | 46.03±0.69 | 45.40±0.53 | 10.54±0.23 |
| | +DC | 48.23±0.51 | 47.40±0.60 | 47.20±0.40 | 46.77±0.58 | 10.98±0.08 |
| | +Var-IC | 53.63±0.89 | 52.80±0.53 | 49.70±1.00 | 45.40±2.07 | 10.85±0.09 |
| | **+Ours** | **55.37±0.68** | **52.07±0.94** | **50.40±0.86** | **48.87±0.94** | **6.35±0.01** |
| Average | Vanilla | 52.96±0.48 | 50.61±0.93 | 46.91±0.88 | 44.60±1.12 | - |
| | +CC | 48.50±0.51 | 48.16±1.07 | 46.37±0.83 | 46.00±0.92 | 10.61±0.29 |
| | +DC | 48.74±0.90 | 47.89±0.84 | 47.77±0.83 | 46.89±0.94 | 10.71±0.23 |
| | +Var-IC | 53.31±0.70 | 51.88±1.26 | 46.52±1.03 | 43.58±1.50 | 10.68±0.25 |
| | **+Ours** | 53.00±0.64 | **51.72±0.90** | **49.56±1.20** | **48.86±1.53** | **6.35±0.15** |

Table 13: Average test accuracy (%) and computational cost (hours) with standard deviations (three runs) on Amazon under different imbalance ratios. The bold indicates the improved results by integrating our method. The *Vanilla* refers to the existing selection methods.

| Dataset | Method | Imbalanced Ratios 1 | 10 | 50 | 100 | Time |
|---------|--------|------|------|------|------|------|
| Amazon | Random | 43.67±0.44 | 38.70±0.60 | 36.27±0.91 | 35.57±0.09 | - |
| | +CC | 41.97±0.56 | 42.97±0.84 | 41.77±1.56 | 42.93±1.84 | 6.35±0.14 |
| | +DC | 41.43±2.02 | 39.07±0.89 | 41.73±0.96 | 41.17±0.82 | 6.31±0.04 |
| | +Var-IC | 42.70±0.00 | 39.33±0.69 | 37.97±0.69 | 37.10±0.80 | 6.51±0.04 |
| | **+Ours** | 43.00±0.60 | **43.60±1.47** | **42.93±0.56** | **43.33±1.24** | **3.90±0.12** |
| | TopK | 47.60±0.00 | 43.20±0.27 | 38.07±1.29 | 37.17±1.36 | - |
| | +CC | 45.27±1.36 | 44.10±2.47 | 42.40±1.33 | 41.97±0.82 | 6.54±0.11 |
| | +DC | 44.77±0.78 | 43.17±0.89 | 43.10±0.67 | 42.43±1.11 | 6.53±0.14 |
| | +Var-IC | 47.2±0.67 | 43.80±0.87 | 39.17±1.69 | 38.40±0.93 | 6.67±0.03 |
| | **+Ours** | 47.30±0.40 | **46.57±0.58** | **43.60±0.27** | 40.87±1.22 | **3.84±0.06** |
| | DPP | 48.10±0.00 | 43.37±0.18 | 39.27±0.56 | 37.77±0.58 | - |
| | +CC | 45.90±0.93 | 44.43±1.36 | 42.60±1.60 | 41.30±0.13 | 6.51±0.08 |
| | +DC | 45.47±0.18 | 43.47±0.98 | 42.70±1.67 | 42.50±0.53 | 6.46±0.16 |
| | +Var-IC | 47.60±0.47 | 44.03±1.18 | 40.97±0.58 | 38.83±0.49 | 6.42±0.02 |
| | **+Ours** | 47.97±0.09 | **45.47±0.44** | **42.13±0.96** | 40.23±0.44 | **3.89±0.10** |
| | VoteK | 48.10±0.00 | 43.37±0.18 | 39.27±0.56 | 37.77±0.58 | - |
| | +CC | 46.10±1.33 | 43.83±0.58 | 42.40±1.33 | 42.30±0.60 | 6.40±0.07 |
| | +DC | 45.13±0.36 | 43.50±1.00 | 42.43±0.38 | 42.10±0.73 | 6.41±0.11 |
| | +Var-IC | 47.73±0.36 | 44.33±0.91 | 39.17±1.69 | 39.07±0.71 | 6.49±0.10 |
| | **+Ours** | 47.93±0.16 | **44.83±0.58** | **43.17±0.69** | 40.73±0.64 | **3.85±0.04** |
| | ConE | 48.10±0.07 | 42.77±0.16 | 37.87±0.76 | 36.33±0.91 | - |
| | +CC | 45.87±0.84 | 43.83±0.58 | 42.40±1.33 | 42.63±0.38 | 6.38±0.06 |
| | +DC | 45.07±0.71 | 43.37±1.58 | 42.50±0.93 | 41.87±0.42 | 6.43±0.20 |
| | +Var-IC | 47.87±0.69 | 44.33±0.91 | 39.17±1.69 | 39.07±0.71 | 6.53±0.14 |
| | **+Ours** | 48.00±0.13 | **44.57±0.84** | **42.93±0.62** | 39.87±0.42 | **3.85±0.07** |
| | ByCS | 48.23±0.09 | 43.43±0.11 | 38.60±1.00 | 37.43±0.71 | - |
| | +CC | 46.40±0.53 | 43.67±0.31 | 42.07±1.11 | 41.93±0.49 | 6.43±0.27 |
| | +DC | 45.37±0.58 | 43.67±1.64 | 42.43±0.98 | 42.27±0.58 | 6.54±0.13 |
| | +Var-IC | 48.20±0.67 | 44.37±0.62 | 39.30±1.73 | 39.67±0.76 | 6.48±0.13 |
| | **+Ours** | 48.03±0.24 | **45.47±0.96** | **42.67±0.24** | 40.57±0.51 | **3.86±0.02** |
| Average | Vanilla | 47.29±0.11 | 42.49±0.31 | 37.94±0.86 | 36.83±0.77 | - |
| | +CC | 45.25±0.93 | 43.81±1.02 | 42.27±1.38 | 42.18±0.71 | 6.43±0.12 |
| | +DC | 44.54±0.77 | 42.71±1.16 | 42.48±0.93 | 42.06±0.70 | 6.45±0.13 |
| | +Var-IC | 46.88±0.47 | 43.37±0.86 | 39.29±1.34 | 38.69±0.73 | 6.52±0.08 |
| | **+Ours** | 47.04±0.27 | **45.08±0.81** | **42.91±0.56** | 40.93±0.75 | **3.87±0.07** |

Table 14: Average test accuracy (%) and computational cost (hours) with standard deviations (three runs) across on Yelp under different imbalance ratios. The bold indicates the improved results by integrating our method. The *Vanilla* refers to the existing selection methods.

| Dataset | Method | Imbalanced Ratios | | | | Time |
| | | 1 | 10 | 50 | 100 | |
| --- | --- | --- | --- | --- | --- | --- |
| Yelp | Random | 44.27±0.36 | 42.10±0.67 | 38.90±0.27 | 38.17±0.22 | - |
| | +CC | 40.93±0.49 | 41.07±0.18 | 40.83±0.29 | 40.37±0.78 | 6.27±0.14 |
| | +DC | 44.27±1.09 | 42.83±2.11 | 42.33±1.78 | 43.50±2.00 | 6.15±0.08 |
| | +Var-IC | 44.20±0.33 | 41.67±0.49 | 40.03±0.71 | 36.83±0.89 | 6.07±0.10 |
| | **+Ours** | **45.40±0.60** | **45.63±0.36** | **44.40±1.73** | **43.30±1.40** | **3.73±0.05** |
| | TopK | 49.70±0.00 | 46.97±0.38 | 43.30±0.27 | 42.17±0.49 | - |
| | +CC | 41.5±0.27 | 42.40±0.53 | 41.63±0.09 | 40.67±0.29 | 5.91±0.13 |
| | +DC | 47.13±0.89 | 45.87±2.22 | 43.83±0.71 | 42.70±0.40 | 5.97±0.32 |
| | +Var-IC | 48.30±0.20 | 45.67±0.76 | 42.27±1.11 | 41.30±0.67 | 6.12±0.25 |
| | **+Ours** | 48.10±0.00 | 46.60±0.40 | **44.93±0.11** | **43.83±0.36** | **3.65±0.10** |
| | DPP | 48.10±0.00 | 46.33±0.89 | 43.90±0.13 | 42.77±0.18 | - |
| | +CC | 41.80±0.13 | 42.43±0.51 | 42.30±0.40 | 41.80±0.47 | 6.25±0.05 |
| | +DC | 47.77±0.64 | 45.87±1.11 | 43.90±0.60 | 42.87±0.69 | 6.21±0.15 |
| | +Var-IC | 48.97±0.69 | 45.27±0.36 | 43.27±0.44 | 41.97±0.44 | 6.21±0.25 |
| | **+Ours** | 48.13±0.04 | 45.83±0.78 | **44.20±0.47** | **44.03±0.49** | **3.66±0.20** |
| | VoteK | 49.20±0.00 | 46.70±0.47 | 43.40±0.53 | 42.07±0.56 | - |
| | +CC | 42.47±0.31 | 42.70±0.67 | 42.97±0.49 | 42.63±0.18 | 6.25±0.06 |
| | +DC | 47.93±0.76 | 45.67±0.71 | 43.70±0.47 | 42.47±0.49 | 6.02±0.44 |
| | +Var-IC | 49.17±0.44 | 45.67±0.76 | 43.60±0.67 | 42.20±0.13 | 6.13±0.14 |
| | **+Ours** | 49.00±0.13 | 46.73±0.16 | **44.10±0.53** | **43.17±0.58** | **3.69±0.21** |
| | ConE | 48.63±0.04 | 46.40±0.53 | 42.73±0.56 | 41.87±0.29 | - |
| | +CC | 42.10±0.53 | 42.83±0.56 | 42.70±0.40 | 41.90±0.27 | 6.10±0.35 |
| | +DC | 47.40±0.40 | 45.17±0.71 | 43.17±0.89 | 42.63±0.84 | 6.07±0.39 |
| | +Var-IC | 48.17±0.56 | 45.10±0.67 | 42.77±0.56 | 42.07±0.18 | 6.03±0.31 |
| | **+Ours** | 48.43±0.11 | **46.57±0.42** | **43.23±0.22** | 42.53±0.38 | **3.75±0.18** |
| | ByCS | 48.97±0.09 | 46.37±0.36 | 43.07±0.11 | 42.00±0.33 | - |
| | +CC | 42.60±0.93 | 43.40±0.93 | 43.13±0.22 | 42.27±0.16 | 6.14±0.09 |
| | +DC | 48.17±0.29 | 45.57±0.84 | 43.10±0.60 | 42.83±0.78 | 5.98±0.10 |
| | +Var-IC | 47.83±0.22 | 45.57±0.62 | 43.53±0.09 | 41.57±0.36 | 6.12±0.26 |
| | **+Ours** | 48.37±0.09 | 46.33±0.24 | **43.57±0.29** | **43.57±0.29** | **3.81±0.05** |
| Average | Vanilla | 48.14±0.08 | 45.81±0.55 | 42.55±0.31 | 41.51±0.34 | - |
| | +CC | 41.9±0.44 | 42.47±0.56 | 42.26±0.31 | 41.61±0.36 | 6.08±0.35 |
| | +DC | 47.11±0.68 | 45.16±1.29 | 43.34±0.84 | 42.83±0.87 | 6.15±0.03 |
| | +Var-IC | 47.77±0.41 | 44.82±0.61 | 42.58±0.6 | 40.99±0.44 | 6.30±0.18 |
| | **+Ours** | 47.91±0.16 | **46.28±0.39** | **44.07±0.56** | **43.46±0.58** | **3.76±0.22** |

Table 15: The statistics, split and evaluation metrics of each dataset.

| Data | Train Set | test dataset | Classes | Evaluation |
| --- | --- | --- | --- | --- |
| Amazon | 25000 | 1000 | 5 | Accuracy |
| AgNews | 20000 | 1000 | 4 | Accuracy |
| Yelp | 25000 | 1000 | 5 | Accuracy |
| Yahoo | 50000 | 500 | 10 | Accuracy |
| Emotion | 15758 | 1974 | 6 | Macro-F1 |
| Natural Question | 25000 | 500 | 5 | Exact Match |
| CodeSearchNet | 18000 | 500 | 6 | Rouge |

Table 16: The instructions, inference templates and example cases of tasks.

| Dataset | Prompt | Example |
|---|---|---|
| Amazon | Task Instruction: Sentiment of the sentence
Inference Verbalizer: Great, Good, Okay, Bad, Terrible?
Input: *Question*
Output: *Answer* | Task Instruction: Sentiment of the sentence
Inference Verbalizer: Great, Good, Okay, Bad, Terrible?
Input: why give me a date for a month then when its suppose to ship,
its running late
Output: Terrible |
| AgNews | Task Instruction: Text Classification Task
Inference Verbalizer: World, Sports, Business or Science New Topic?
Input: *Question*
Output: *Answer* | Task Instruction: Text Classification Task
Inference Verbalizer: World, Sports, Business or Science New Topic?
Input: EBay Buys 25 Percent Stake in Craigslist Network By MAY WONG
SAN JOSE, Calif. (AP) – Online auctioneer eBay Inc.
Output: Science |
| Yelp | Task Instruction: Sentiment of the sentence
Inference Verbalizer: Great, Good, Okay, Bad, Terrible?
Input: *Question*
Output: *Answer* | Task Instruction: Sentiment of the sentence
Inference Verbalizer: Great, Good, Okay, Bad, Terrible?
Input: Awesome place!!! You must go and try all the services!!!!
Output: Good |
| Yahoo | Topic of the text:
Society & Culture, Science & Mathematics,
Health, Education & Reference, Computers & Internet,
Sports, Business & Finance, Entertainment & Music,
Family & Relationships, Politics & Government?
Input: *Question*
Output: *Answer* | Topic of the text: Society & Culture, Science & Mathematics,
Health, Education & Reference, Computers & Internet,
Sports, Business & Finance, Entertainment & Music,
Family & Relationships, Politics & Government?
Input: what is god's kingdom that we are told to pray for?
Output: Computers & Internet |
| Emotion | Task Instruction: Sentiment of the sentence
Inference Verbalizer: Sadness, Joy, Love, Anger, Fear, Surprise?
Input: *Question* Output: *Answer* | Task Instruction: Sentiment of the sentence Inference
Verbalizer: Sadness, Joy, Love, Anger, Fear, Surprise?
Input: *im feeling generous this week*
Output: *Joy* |
| NQ | Question: *Question*
Answer: *Answer* | Question: *who is the CEO of what's up*
Answer: *Jan Koum* |
| CodeSearchNet | Summarize the code.
Input: *Input*
Output: *Output* | Summarize the code.
Input: "func NewMessage() Message {
return Message{
Context: context.Background(),
Headers: map[string]string{},
Data: render.Data{},
moot: &sync.RWMutex,}}"
Output: *NewMessage builds a new message.* |

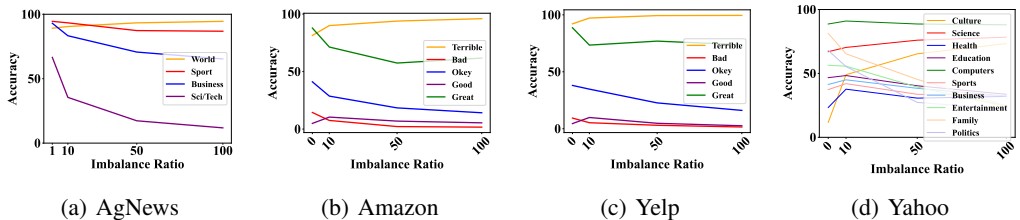

Figure 5: Average accuracy for each class across six selection methods in AgNews, Amazon, Yelp and Yahoo datasets across various imbalance ratios.

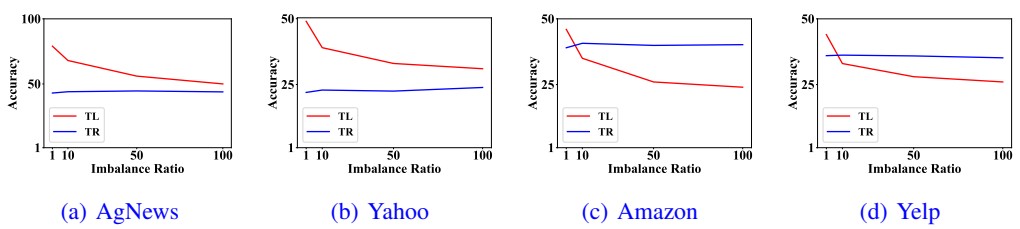

Figure 6: Averaged accuracy of Task Recognition (TR) and Task Learning (TL) across six selection methods on the AgNews, Amazon, Yelp, and Yahoo datasets under various imbalance ratios.

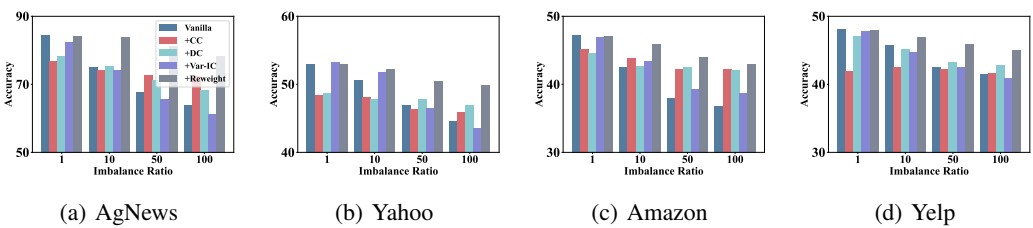

Figure 7: The performance of ICL across different datasets when the scoring function is reweighted using the optimal importance factors $w$.

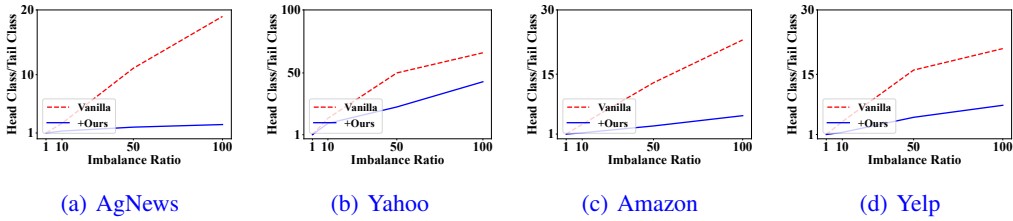

Figure 8: Ratio of head to tail classes in the demonstration set across across six selection methods on four classification tasks with imbalanced ratios.

