# OpenReview forum: "Exploring Imbalanced Annotations for Effective In-Context Learning"
_ICLR.cc/2026/Conference — Submitted to ICLR 2026_

### Official Review · Reviewer_yh6B · 2025-10-31

**Soundness:** 3
**Presentation:** 2
**Contribution:** 2
**Rating:** 2
**Confidence:** 3

**Summary:**

This paper investigates the performance degradation of In-Context Learning when demonstration examples are selected from class-imbalanced annotated datasets. The authors show this issue persists despite varying selection methods, or calibration techniques. They propose Reweighting with Importance Factors (RIF), a method that estimates class weights using a balanced subset and Bayesian optimization. During inference, RIF re-weights the scores used by standard selection methods (e.g., TopK) to counteract the imbalance.

**Strengths:**

1. Addresses the important and practical problem of performance degradation in In-Context Learning due to class imbalance in the annotated dataset, a real-world scenario.
2. Proposes RIF, a method involving 'pre-hoc' score re-weighting based on importance factors estimated via Bayesian optimization on a balanced subset, conceptually contrasting with 'post-hoc' calibration.
3. RIF is technically sound and demonstrates consistent empirical improvements over vanilla selection and calibration baselines across extensive experiments (multiple datasets, imbalance ratios, LLMs).
4. Shows better computational efficiency compared to calibration methods.

**Weaknesses:**

1. Further Validation Suggested for Conceptual Advantages: The paper compellingly argues that RIF's advantage over undersampling lies in preserving valuable information, potentially leading to better generalization and robustness. To further solidify this key argument, the paper would benefit significantly from direct experimental validation comparing RIF against undersampling on metrics related to generalization (e.g., out-of-distribution performance) or robustness.
2. Opportunity for Broader Baseline Comparison: To provide a more comprehensive picture of RIF's performance relative to alternative strategies, the authors might consider including comparisons against potentially strong baselines that combine existing techniques. For example, evaluating RIF against hybrid approaches, such as undersampling and calibration, could further contextualize its advantages and provide a more rigorous assessment.
3. Clarifying Positioning Relative to Prior Work: While the RIF mechanism itself is novel, the paper's positioning as the "first systematic study" on "Imbalanced Annotation" affecting selection could be enhanced for clarity. Discussing how RIF specifically differs from recent related works [1,2] that also consider class imbalance in the annotated dataset for ICL demonstration selection would help readers better appreciate its unique contributions against the current research landscape.
4. Enhancing Clarity on Hyperparameter K': The results indicate performance sensitivity to the candidate size K' (Figure 2b) To aid reproducibility and practical application, the paper could be enhanced by explicitly reporting the K' value used for the main results (Table 1). Additionally, offering practical guidance or analysis on selecting K' considering the trade-off between performance and computational cost, would be valuable for readers.
5. Minor Presentation Flaws: There are inconsistencies (e.g., ByDC vs. ByCS typo), missing visualizations (e.g., RIF's direct impact on Head/Tail ratio in selection), and a lack of precision in describing the imbalanced dataset generation process.

[1] Strategic Demonstration Selection for Improved Fairness in LLM In-Context Learning (EMNLP’24)
[2] IM-Context: In-Context Learning for Imbalanced Regression Tasks (TMLR’24)

**Questions:**

1. Clarification on Table 1 and the Implication of phi=1 Performance: Does it evaluate performance on the same (balanced) test set while varying the imbalance ratio (phi) of only the annotation dataset used for demonstration selection? If this understanding is correct, the results consistently show that using an annotation dataset balanced via undersampling (phi=1) achieves the highest ICL performance across all scenarios. Doesn't this finding suggest that simply undersampling a real-world imbalanced dataset to achieve balance might be the most effective strategy for maximizing ICL performance on these benchmarks, potentially more so than applying RIF to the imbalanced dataset?
2. Comparison to Hybrid/Stronger Baselines: Could you comment on the potential performance of hybrid baselines like undersampling with calibration or augmentation with oversampling?
3. Understanding RIF's Distinctions from Related Work: How does RIF's contribution specifically differ from recent ICL selection methods [1,2] that address data imbalance implicitly while focusing on fairness (e.g., prioritizing underrepresented groups), or imbalanced regression?
4. Clarification on K' Value and Guidance: What specific value of K' was used to generate the main results presented in Table 1? Could you provide practical guidance or heuristics for setting K' based on dataset characteristics (e.g., imbalance ratio, tail class size) and computational constraints? Furthermore, what is the empirical computational overhead (e.g., latency increase) observed when significantly increasing K' (e.g., from 100 to 1600 in your Fig 2b experiments)?
5. Imbalanced Dataset Generation Details: Could you please provide the precise mathematical function or procedure used to determine the number of samples for intermediate classes when creating the synthetic imbalanced datasets with phi=10, 50, 100?

---

> ### Author Response · Authors · 2025-11-25
> **Response to Reviewer yh6B**
>
> Thank you for the constructive and elaborate feedback. Please find our response below.
>
> **1. Comparison with rebalancing method [W1, Q1]**
>
> Sorry for the confusion. **Appendix G.1** presents the experimental results of classicial rebalancing methods, including oversampling, undersampling, stratified sampling and re-weighting. The table G.1 reports the average test accuracy across six selection methods on the AgNews and Yahoo datasets with different imbalance ratios. The results show that the **rebalancing methods yield only marginal improvements under imbalanced annotations**, and our method consistently outperforms them. To avoid any potential confusion, we have relocated the empirical results to subsection 3.2 of the main paper.
>
> **2. Evaluation on boarder baselines [W2, Q2]**
>
> Thank you for the question. Here, we added boader baselines, including FCG [5], over-sampling+CC,  under-sampling+CC and stratified-sampling+CC. The table below presents average test accuracy of the baselines and our method on the two classification tasks: AgNews and Yahoo. The results show that **our method can outperform these ensemble baselines**. We added the results in Appendix G.2 of final version.
>
> |Imbalance Ratios|1|10|50|100|
> |-|-|-|-|-|
> |Vanilla|66.57|62.29|55.95|53.78|
> |+FCG|63.34|62.64|61.25|60.50|
> |+Over-sampling+CC|62.71|60.44|59.02|58.65|
> |+Under-sampling+CC|62.71|60.72|59.68|59.28|
> |+Stratified-sampling+CC|61.30|61.20|60.88|60.27|
> |**+Ours**|66.72|**65.94**|**63.14**|**62.22**|
>
>
> **3. Concerns of related works [W3]**
>
> Thank you for the suggestion. The ICL are closely related to the concept of label bias [1], where language models are biased toward certain answers during in-context learning. In particular, the most relevant part is the majority label bias, a type of context label bias that leads the model to predict answers that appear frequently in the prompt [2-3].  An empirical work [4] suggested that label imbalance does not substantially impact ICL performance in binary classification. The subsequent works [2-3] delved deeper into this challenge and proposed to address it by calibrating the model's output probabilities to compensate for this bias. Additionally, other works [5-6] improve the fairness of ICL across classes by increasing the proportion of minority examples in the demonstration set. Besides, some study [7] discusses how in-context learning emerges when the pre-training data exhibits particular distributional properties. **While prior studies have typically focused on imbalanced distributions within prompts, our work shifts the emphasis to class imbalance in the annotation datasets from which demonstrations are drawn**. Notably, our work presents the first systematic attempt to tackle the class-imbalanced issue from the perspective of demonstration selection. We incorporated this discussion into Appendix C of the revised version and would greatly appreciate it if you could point us to any related works we might have overlooked.
>
>
>
>
> **4. Analysis of selected candidate sizes $K'$ [W4, Q4]**
>
> Thank you for the questions. Please find the detailed analysis in part 2 of the General Response.
>
>
>
>
> **5. Details of imbalance generation [W5, Q5]**
>
> Thank you for pointing out the potential confusion. We construct imbalanced datasets following a standard exponential distribution. Specifically, for each class $i$, the number of examples is given by $N_i=(\frac{N}{k})\times$ $(\frac{1}{\phi})^{\frac{i}{k-1}}$, where $k$ is total number of classes, $N$ is total number of annotated dataset and $\phi$ denotes the imbalance ratio. We added the details in Line 366 of revised version.
>
>
> **6. Concerns of typo and visualization [W5]**
>
> Thank you for pointing out the issues of typo and visualization. We reply to each question below.
> * In all tables and figures in this paper, we replaced the name "ByDC" with "ByCS".
> * Figure 8 shows ratio of head to tail classes in the demonstration set selected by existing selection methods and our method. Clearly, our method prevents over-selection from dominant classes and achieves class distributions in the demonstration set that are similar to those of the balanced annotated dataset.
>
> To avoid any potential confusion, we fixed these typos and added the visualization  in the revised version.
>
>
> [1] Language models are few-shot learners. NeurIPS 2020.
>
> [2] Calibrate before use: Improving few-shot performance of language models. ICML 2021.
>
> [3] Mitigating label biases for in-context learning. ACL 2023.
>
> [4] Investigating the learning behaviour of in-context learning: a comparison with supervised learning. ECAI 2023.
>
> [5] Strategic Demonstration Selection for Improved Fairness in LLM In-Context Learning. EMNLP 2024.
>
> [6] IM-Context: In-Context Learning for Imbalanced Regression Tasks. TMLR 2024.
>
> [7] Data Distributional Properties Drive Emergent In-Context Learning in Transformers. NeurIPS 2022.

---

### Official Review · Reviewer_Xduo · 2025-10-31

**Soundness:** 2
**Presentation:** 2
**Contribution:** 2
**Rating:** 2
**Confidence:** 4

**Summary:**

This paper mainly focuses on the class imbalance in the demonstration selection of the conventional in-context learning paradigm. In this paper, the authors demonstrate that the class distribution degrades the performance of ICL regardless of the selection methods and the classical rebalancing methods, which focus solely on class weights. Meanwhile, the conventional rebalancing weights tend to yield poor performance. Thus, a method, Reweighting with Importance Factors (RIF) is proposed to solve this problem. According to the empirical results, the RIF method achieves good performance on common benchmarks.

**Strengths:**

- This paper explores the class imbalance problem in the demonstration selection of in-context learning, which is useful for both academic and industrial communities.
- This paper proposes a simple method, RIF, to solve the problem.
- The results reported in the paper look good.

**Weaknesses:**

- The motivation of this paper is a little bit trivial. I think it would be common sense that the imbalanced data can result in degradation of performance. To enhance the paper, it would be better to provide more insights regarding this problem.

- The formulations in this paper have many problems.

**Questions:**

- The equation in Remark 1 seems to assume that the generation of $\boldsymbol{{\rm y}}$ does not rely on the in-context demonstrations? How do you obtain such a conclusion?

- From both Remark 1 and Eq. (3), it seems that only the distribution of demonstrations is considered. However, the prior knowledge embedded in the LLM is also essential. In this paper, the prior knowledge is not taken into consideration.

- Could you please provide more explanation regarding your estimate in Section 3.2? Specifically, how do you measure the probability? How do you measure the average probability?

- In the equation in Line 238-239, the notions $\boldsymbol{{\rm x}}$ and $\boldsymbol{{\rm y}}$ do not appear in $M(\cdot, \cdot)$.

---

> ### Author Response · Authors · 2025-11-25
> **Response to Reviewer Xduo**
>
> Thank you for the positive and constructive feedback. Please find our response below:
>
> **1. In-depth analysis of imbalanced annotations [W1]**
>
> Thank you for the suggestion. Please find the detailed analysis in part 1 of the General Response.
>
> **2. Clarification of Remark 1 [Q1, Q2]**
>
> Sorry for the confusion. We respond to these questions point by point below.
> * **The generation of $y$ does not rely on the in-context demonstrations?** We restate Theorem 1 from the prior study [1] as Remark 1 which introduces how ICL makes predictions through the given in-context demonstrations. According to Remark 1, ICL can be expressed as $f_\theta (C_K,x_t)\approx \arg\max_y P_c (y|x_t)$. ICL enables an LLM to learn the latent demonstration parameter from the in-context demonstrations $C_K$ and generate output $y$ according to the demonstration distribution $P_c(x,y)$.
> * **The prior knowledge is not taken into consideration.** The pretrained LLM’s prior knowledge is already encoded in the implicit prior $p(\theta)$ and likelihood $p(y∣x,\theta)$. Our work mainly examines how the class distribution of the demonstration set affects the posterior inference over $\theta$, while keeping the pretrained prior fixed. Therefore, the prior knowledge is not neglected but rather treated as an immutable component of the model.
>
>
>
>
>
>
> **3. Details of calibration methods [Q3]**
>
> Sorry for the confusion. We respond to these questions point by point below.
> * **How do you measure the probability?** Following the previous studies [2-3], we use the LLM’s perplexity on the sequence formed by concatenating the input with each candidate answer as a proxy for its probability. Specifically, for each possible label, we concatenate each tokenized input-output pair $(x,y)$ and obtain the corresponding tokenized sequence $c=(x_1,...,x_{|x|},y_1,...,y_{|y|})$ where $|c|=|x|+|y|$. The perplexity of $c$ is calculated as $Perplexity(c)=\exp\{-\frac{1}{|c|}\sum_{i=1}^{|c|}\log P_\theta (c_i|c_{<i}) \}$ where $\log P_\theta (c_i|c_{<i})$ is the log-likelihood of the $i$-th token conditioned on the preceding tokens $c_{<i}$.
> * **How do you measure the average probability?** For each candidate answer, we replace the test input with multiple context-free words or sentences, compute a series of perplexity values conditioned on the given demonstrations $C_K$, and then take their average.
>
>
> **4. Concerns of notions [W2, Q4]**
>
> Thank you for pointing out the issues of notation. We have fixed these in the revised version.
>
>
> [1] An explanation of in-context learning as implicit bayesian inference. ICLR 2022.
>
> [2] OpenICL: An open-source framework for in-context learning. ACL 2023.
>
> [3] Revisiting demonstration selection strategies in in-context learning. ACL 2024.

---

### Official Review · Reviewer_Cf7b · 2025-10-31

**Soundness:** 2
**Presentation:** 3
**Contribution:** 2
**Rating:** 4
**Confidence:** 4

**Summary:**

This paper investigates how class imbalance in annotated datasets affects the performance of in-context learning (ICL) in large language models. The authors demonstrate empirically that imbalanced annotations degrade ICL performance and propose a reweighting approach, Reweighting with Importance Factors (RIF), that adjusts the demonstration selection process to mitigate the imbalance. RIF is evaluated across multiple benchmarks and models, showing modest but consistent performance improvements.

**Strengths:**

* Clarity and Presentation: The paper is well-written and clearly structured. The motivation, methodology, and results are presented in an organized and easy-to-follow manner.
* Comprehensive Evaluation: The experiments are extensive, covering multiple datasets, imbalance ratios, and both open-weight and API-based models.
* Simplicity of the Proposed Method: RIF is straightforward to implement and integrates easily with existing demonstration selection methods.

**Weaknesses:**

* Lack of Novelty: The core idea—reweighting samples based on estimated importance factors—is not fundamentally new and aligns closely with well-known class balancing and importance sampling techniques. The contribution seems more incremental than conceptual.
* Limited Theoretical Depth: The theoretical justification is fairly standard, and the paper mostly reiterates known results in imbalance learning, adapted to the ICL setting.
* Modest Empirical Gains: While improvements are consistent, they are relatively small (typically 3–5%), and it is not entirely clear whether such gains justify a new method.
* Framing: The paper positions itself as the “first” to study imbalance in ICL, but this framing feels overstated given prior works that already examine selection biases and label imbalance in few-shot or ICL setups.

**Questions:**

Novelty of RIF:

* The proposed Reweighting with Importance Factors (RIF) seems closely related to standard class reweighting and importance sampling methods widely used in imbalance learning. Could the authors clarify what is fundamentally novel in RIF beyond adapting these existing ideas to the ICL setting?

Effectiveness vs. Simple Baselines:

* Have the authors compared RIF to simpler baselines, such as class-balanced sampling or uniform per-class Top-K selection? Without such baselines, it’s unclear whether the observed gains come from the reweighting mechanism itself or simply from balancing the demonstration set.

Estimation of Importance Factors:

* The paper mentions estimating importance factors via Bayesian optimization on a balanced subset. Why was this approach chosen over more straightforward estimations (e.g., using normalized class frequencies)? How sensitive is the method to the subset size or its sampling variability?

Practical Applicability:

* RIF assumes access to a balanced subset for estimating class weights. In real-world long-tailed or low-resource scenarios, tail classes may have very few examples. How practical is this assumption, and can RIF still be applied effectively when such a balanced subset is infeasible to obtain?

---

> ### Author Response · Authors · 2025-11-25
> **Response to Reviewer Cf7b**
>
> Thank you for the positive and constructive feedback. Please find our response below:
>
> **1. Comparison with rebalancing method [W1, Q2, Q3]**
>
> Sorry for the confusion. **Appendix G.1** presents the experimental results of classicial rebalancing methods, including oversampling, undersampling, stratified sampling and re-weighting. The table G.1 reports the average test accuracy across six selection methods on the AgNews and Yahoo datasets with different imbalance ratios. The results show that the **rebalancing methods yield only marginal improvements under imbalanced annotations**, and our method consistently outperforms them. To avoid any potential confusion, we have relocated the empirical results to subsection 3.2 of the main paper.
>
> **2. Clarification of our method [W2, Q1]**
>
> Great suggestion. We would like to clarify that classical re-sampling and re-weighting methods in imbalance learning focus solely on the annotated dataset, whereas the distribution of the ICL demonstration set is jointly determined by both the annotated dataset and the demonstration selection strategy. The results in **Appendix G.1** confirm that classical imbalance-learning methods do not transfer directly to the ICL setting. RIF introduces a data-driven correction by reweighting importance factors estimated via Bayesian optimization on a balanced subset. Comprehensive experiments show that RIF prevents over-selection of dominant classes while preserving the effectiveness of existing selection methods for ICL.
>
> **3. Significant improvements with minimal computational overhead [W3]**
>
> Thank you for raising the concern. We would like to clarify that the improvements are significant and achieved with minimal computational overhead. **Table 1** reports average test accuracy and computational cost  with standard deviations (three runs) across six selection methods on four classification datasets with various imbalance ratios. The results show that our method improves ICL performance, and the improvement exceeds the reported standard deviations, thereby demonstrating statistical significance. For example, our method increases the average accuracy of Vanilla ICL from $46.47\pm0.88$ to $52.07\pm0.98$, yielding an absolute improvement of 5.60 percentage points. Furthermore, compared with existing calibration methods, our approach achieves an additional 2–5 accuracy improvement while requiring only 50% of their computational cost.
>
> **4. Concerns of related works [W4]**
>
> Thank you for the suggestion. The ICL are closely related to the concept of label bias [1], where language models are biased toward certain answers during in-context learning. In particular, the most relevant part is the majority label bias, a type of context label bias that leads the model to predict answers that appear frequently in the prompt [2-3].  An empirical work [4] suggested that label imbalance does not substantially impact ICL performance in binary classification. The subsequent works [2-3] delved deeper into this challenge and proposed to address it by calibrating the model's output probabilities to compensate for this bias. Additionally, other works [5-6] improve the fairness of ICL across classes by increasing the proportion of minority examples in the demonstration set. Besides, some study [7] discusses how in-context learning emerges when the pre-training data exhibits particular distributional properties. **While prior studies have typically focused on imbalanced distributions within prompts, our work shifts the emphasis to class imbalance in the annotation datasets from which demonstrations are drawn**. Notably, our work presents the first systematic attempt to tackle the class-imbalanced issue from the perspective of demonstration selection. We incorporated this discussion into Appendix C of the revised version and would greatly appreciate it if you could point us to any related works we might have overlooked.
>
>
>
>
> **5. Analysis of selected candidate sizes $K'$ [Q3]**
>
> Thank you for the questions. Please find the detailed analysis in part 2 of the General Response.
>
>
>
> [1] Language models are few-shot learners. NeurIPS 2020.
>
> [2] Calibrate before use: Improving few-shot performance of language models. ICML 2021.
>
> [3] Mitigating label biases for in-context learning. ACL 2023.
>
> [4] Investigating the learning behaviour of in-context learning: a comparison with supervised learning. ECAI 2023.
>
> [5] Strategic Demonstration Selection for Improved Fairness in LLM In-Context Learning. EMNLP 2024.
>
> [6] IM-Context: In-Context Learning for Imbalanced Regression Tasks. TMLR 2024.
>
> [7] Data Distributional Properties Drive Emergent In-Context Learning in Transformers. NeurIPS 2022.

---

> > ### Author Response · Authors · 2025-11-25
> > **Response to Reviewer Cf7b**
> >
> > **6. Results of different selection methods [Q3]**
> >
> > Sorry for the confusion. **Tables 14-17** present the average test accuracy of six selection methods on four classification datasets with different imbalance ratios. The results demonstrate that **our method is algorithm-agnostic** and consistently enhances ICL performance across diverse selection methods.
> >
> > **7. Results of extreme imbalanced ratios [Q4]**
> >
> > Great suggestion. We conducted a new experiment on dataset with extreme imbalanced ratios. We present the results in part 2 of the General Response.

---

### Official Review · Reviewer_SG7B · 2025-10-31

**Soundness:** 2
**Presentation:** 2
**Contribution:** 1
**Rating:** 2
**Confidence:** 5

**Summary:**

This paper investigates how class imbalance in annotation datasets affects in-context learning (ICL) performance. The authors show that imbalanced annotations significantly degrade ICL across various tasks, and that existing calibration methods fail to address this. They propose RIF (Reweighting with Importance Factors), which estimates importance weights using a balanced subset and reweights scoring functions during demonstration selection. Experiments across different datasets show incremental improvement with accuracy gains of up to 5.6% on highly imbalanced datasets.

**Strengths:**

S1: The paper is well-executed experimentally, with comprehensive evaluations across multiple models (OPT, LLaMA, ChatGPT, Gemini), datasets, and imbalance ratios. The experimental protocol is solid and well explained. The explanation and figures are straightforward to understand.

S2: The finding that a larger dataset doesn't automatically address the imbalance issue is useful.

**Weaknesses:**

W1: This is my main concern. The problem and solution seems expected and unsurprising. It isn't surprising that class imbalance harms ICL performance. The solution to use a balanced subset to calculate a weighting factor is also hardly novel or particularly interesting. It is unclear why not just used the balanced set for ICL?

W2: The novelty seem limited. Importance sampling and reweighting for class imbalance are well known techniques addressing imbalance.

W3: The improvements are incremental and in many cases barely above noise. Looking at Table 1, many improvements are within 2% on top of a ~50% baseline. It is hard to conclude how impactful these improvements will be.

W4: It is unclear how these LLMs are selected. They seem to span old models to recent models without any clear reason or selection criterion.

**Questions:**

Q: It is unclear if this ICL is in the learning regime or retrieval regime please see: Pan et al: https://arxiv.org/abs/2305.09731

---

> ### Author Response · Authors · 2025-11-25
> **Response to Reviewer SG7B**
>
> Thank you for the constructive and elaborate feedback. Please find our response below.
>
> **1. Comparison with rebalancing method [W1, W2]**
>
> Sorry for the confusion. **Appendix G.1** presents the experimental results of classicial rebalancing methods, including oversampling, undersampling, stratified sampling and re-weighting. The table G.1 reports the average test accuracy across six selection methods on the AgNews and Yahoo datasets with different imbalance ratios. The results show that the **rebalancing methods yield only marginal improvements under imbalanced annotations**, and our method consistently outperforms them. To avoid any potential confusion, we have relocated the empirical results to subsection 3.2 of the main paper.
>
>
> **2. Significant improvements with minimal computational overhead [W3]**
>
> Thank you for raising the concern. We would like to clarify that the improvements are significant and achieved with minimal computational overhead. **Table 1** reports average test accuracy and computational cost  with standard deviations (three runs) across six selection methods on four classification datasets with various imbalance ratios. The results show that our method improves ICL performance, and the improvement exceeds the reported standard deviations, thereby demonstrating statistical significance. For example, our method increases the average accuracy of Vanilla ICL from $46.47\pm0.88$ to $52.07\pm0.98$, yielding an absolute improvement of 5.60 percentage points. Furthermore, compared with existing calibration methods, our approach achieves an additional 2–5 accuracy improvement while requiring only 50% of their computational cost.
>
>
> **3. Clarification of selected LLMs [W4]**
>
> Thank you for the great suggestion. GPT, LLaMA, and OPT are three major LLM families that have been widely used to study ICL in prior work [1–3]. We use OPTs in most experiments due to their diverse range of parameter sizes (e.g., 125M-66B) and faster inference time on a single 4090 GPU. Meanwhile, we also present ablation study of different LLMs, including open-weight models: OPT-6.7B, OPT-13B, OPT-30B, LLAMA-3-8B, and LLAMA-3-70B and APIs: ChatGPT3.5-Turbo and Gemini-2.0-Flash. Appendix G.5 also shows the same phenomenon across different model architectures and sizes: the ICL performance get worse at larger imbalanced ratios, and our method can significantly improve the performance.
>
>
> **4. In-depth analysis of imbalanced annotations [Q1]**
>
> Thank you for the suggestion. Please find the detailed analysis in part 1 of the General Response.
>
>
> [1] Rethinking the role of scale for in-context learning: An interpretability-based case study at 66 billion scale. ACL 2023.
>
> [2] MEND: Meta demonstration distillation for efficient and effective in-context learning. ICRL 2024.
>
> [3] What In-Context Learning "Learns" In-Context: Disentangling Task Recognition and Task Learning. ACL 2023.

---

> > ### Comment · Reviewer_SG7B · 2025-11-28
> >
> > I thank the authors for their detailed response and additional experiments.
> >
> > The paper has no weaknesses in clarity or rigorousness.
> >
> > ---
> >
> > **However, I maintain my original assessment, which was about the scope of the research not about its execution.** The research question being addressed is quite an unsurprising one and more importantly the proposed solution is neither using a new insight or yielding surprising improvements. The contribution feels more like demonstrating that a method which should work can be indeed applied.
> >
> > Unfortunately, I do not think that going over this paper gave me any belief updates or novel ideas to apply to practical problems.
> >
> > I thank the authors again for their reply.

---

### Official Review · Reviewer_S1Zm · 2025-11-01

**Soundness:** 3
**Presentation:** 3
**Contribution:** 2
**Rating:** 6
**Confidence:** 3

**Summary:**

The paper studies how class-imbalanced annotation pools degrade in-context learning (ICL) because selection methods over-pick head classes, biasing priors in prompts, and proposes RIF, a simple reweighting scheme that estimates per-class importance factors from a small balanced subset via Bayesian optimization and then reweights the scoring function (e.g., cosine similarity in TopK) to counteract head dominance during demo selection.

**Strengths:**

- Clear empirical evidence that pool-level class imbalance hurts ICL across selection methods and that common calibration schemes (CC, DC, Var-IC) fail or even neutralize advanced selectors; figures and tables isolate the effect across models and imbalance ratios.​
- Simple, method-agnostic reweighting that plugs into existing selectors without model fine-tuning and only requires model outputs; computational overhead is modest and often lower than calibration baselines in reported settings.​
- Sensible theoretical framing via importance sampling linking expected risk to class-prior mismatch in selected demonstrations, motivating per-class importance factors and reweighted selection distributions.

**Weaknesses:**

- Dependence on building a balanced subset $\mathcal{D}_b$ from tail classes can be impractical when tails are extremely scarce, which the paper acknowledges but does not resolve with alternatives (e.g., generative augmentation, semi-supervised density estimation).​
- Importance-factor estimation uses Bayesian optimization on $\mathcal{D}_b$ with heuristic initialization; sensitivity to surrogate choices, search budget, noise in evaluation, and cross-dataset transferability of learned weights is under-explored beyond limited robustness plots.​
- Theoretical analysis assumes shared conditional $P_c(x|y)=P_t(x|y)$ and reduces imbalance to class priors; distribution shifts in x within a class (domain shift) or long-tail feature skew are not addressed, potentially limiting guarantees in real applications ​.
- Fairness and compute parity: baselines sometimes incur extra calibration passes while RIF reports lower hours; more controlled compute-matched comparisons and wall-time/memory profiles for large-K candidate selection would strengthen efficiency claims.​
- Generation tasks aggregation into coarse “categories” (e.g., NQ) simplifies imbalance as class priors; how RIF extends to open-ended outputs, span-level evaluation, or attribute-level long tails is only preliminarily explored with small gains (e.g., EM +1.7 absolute at 100:1).​
- Reweighting operates at class level; many real-world imbalances are multi-attribute or hierarchical (topic × sentiment × length); the method does not consider multi-label, group, or continuous-attribute imbalance beyond a brief note, risking overcorrection or under-coverage.​
- The approach assumes access to class labels during selection for weight application; for tasks without explicit classes or where labels are latent (retrieval-based reasoning), mapping to “classes” is ad hoc and may inject template bias (Appendix templates).​
- Statistical rigor could improve: many tables show absolute gains but limited hypothesis testing, confidence intervals, or multiple-seed variance in the main text; robustness to prompt formats and verbalizer choices, known to sway ICL, is mostly in appendices or not systematically isolated.​
- Large-K candidate requirement to include tails increases retrieval cost; guidance to set K vs imbalance ratio is qualitative, with no adaptive candidate control or approximate nearest neighbor strategies discussed for scaling.

**Questions:**

- How does RIF perform when tails are extremely small (e.g., ≤5 examples/class)? Can synthetic balancing, weak labels, or external encoders help estimate reliable importance factors without $\mathcal{D}_b$?​
- What is the sensitivity of the learned weights to prompt format, verbalizers, and decoding changes, and can a single set of weights transfer across related tasks or models without re-optimization?​
- Can RIF be extended beyond discrete classes to group, attribute, or continuous-conditioned imbalance (e.g., length, domain, language) by learning importance over clusters or learned partitions?​
- For generation, can importance be defined over answer types or latent intents rather than coarse categories, and does that yield larger gains on EM/ROUGE while avoiding template-induced artifacts?​
- Under domain shift where $P_c(x|y) \neq P_t(x|y)$ can a covariate shift correction be layered with class-prior reweighting, and how would the objective and estimation change practically ​?
- What is the runtime/memory profile as K and class count grow (e.g., K = 2k, |Y| = 100), and can approximate retrieval plus per-class quotas or two-stage filtering retain gains with lower cost

---

> ### Author Response · Authors · 2025-11-25
> **Response to Reviewer S1Zm**
>
> Thank you for the positive and constructive feedback. Please find our response below:
>
> **1. Results of extreme imbalanced ratios [W1, Q1]**
>
> Great suggestion. We conducted a new experiment on dataset with extreme imbalanced ratios. We present the results in part 2 of the General Response.
>
> **2. Analysis of importance factors estimation [W2, W8, Q2]**
>
> Thank you for raising the concern. Here, we provide a detailed analysis of this concern.
>
> * **The learned weights of our method exhibit cross-model transferability.** We conducted a new ablation experiment to verify the cross-model transferability of learned weights. We estimate the importance factors using various LLMs (OPT-2.7B, 6.7B, 13B) and evaluate them on a different model (OPT-6.7B). The results reveal that our method can improve ICL performance using importance factors estimated by different LLMs and demonstrates strong cross-model transferability of the learned weights. We added the experiment in line 453-460 of the revised version.
>
> |Imbalance Ratios|1|10|50|100|
> |-|-|-|-|-|
> |Vanilla|66.57|62.29|55.95|53.78|
> |OPT-2.7B|66.46|65.12|62.01|60.82|
> |OPT-6.7B|66.58|65.70|62.93|61.98|
> |OPT-13B|66.76|65.78|63.22|62.50|
>
> * **Our method works with only a few iterations.** The bayesian optimization employed in our method is well-suited for efficiently optimizing non-differentiable and black-box functions with a few iterations [1-2]. Here, we conducted a new experiment on our method with different iterations (e.g., 10, 30 and 50). The table below presents the average test accuracy across six selection methods on AgNews and Yahoo datasets with different imbalance ratios. The results show that our method work well even with a few iterations (e.g. 10). In fact, we simply set the iterations 30 by default in our experiments. We add the results in Appendix G.3.
>
> |Imbalance Ratios|1|10|50|100|
> |-|-|-|-|-|
> |Vanilla|66.57|62.29|55.95|53.78|
> |Iteration=10|66.24|64.32|60.62|59.48|
> |Iteration=30|66.58|65.70|62.93|61.98|
> |Iteration=50|66.72|65.98|63.34|62.42|
>
> * **Our method is insensitive to different prompt formats.**
> Following the reviewer's advice, we conducted a new exmperiment to estimate importance factors using different prompt formats. The results (vanilla/**+ours**) in the table below present that our method can consistently improve the performance of ICL with various types of prompt formats. This demonstrates that **our method is insensitive to prompt formats.** We add the results in Appendix G.4.
>
> |Imbalance Ratios|1|10|50|100|
> |-|-|-|-|-|
> |Format 1|66.57/66.58|62.29/**65.70**|55.95/**62.93**|53.78/**61.98**|
> |Format 2|67.17/67.13|63.65/**65.52**|58.21/**63.94**|56.33/**63.33**|
> |Format 3|66.97/66.70|62.05/**65.24**|55.51/**62.52**|53.13/**61.33**|
>
> **3. Clarification of conditional distribution [W3, Q5]**
>
> Thank you for raising the concern.  We would like to clarify that RIF estimates importance factors directly through Bayesian optimization on a a balanced subset and doesn't rely on any parametric distributional assumptions. Therefore, any class-conditional feature shift is already reflected in the balanced subset, and RIF automatically adjusts the importance factors accordingly. In practice, this makes RIF applicable even when $P_c(x|y)\neq P_t(x|y)$. We have fixed these in the updated version.
>
>
>
> **4. Analysis of selected candidate sizes $K'$ [W4, W9, Q6]**
>
> Thank you for the questions. Please find the detailed analysis in part 2 of the General Response.
>
> **5. Results of open-ended tasks [W5, Q4]**
>
> Thank you for raising the concern. In the Discussion section (Subsection 6.1), we have presented empirical results on an open-ended task: Code Summarization (CodeSearchNet). The CodeSearchNet dataset contains six programming languages, including Go, Java, JavaScript, PHP, Python, and Ruby. **Figure 3 (b)** demonstrate same pheonomenon as main experiments in the manuscript: the ICL performance of LLMs get worse at larger imbalanced ratios, and our method can significantly improve the performance, especially at large imbalanced ratios (e.g., 100). For example, our method improves the average Rouge Score of CodeSearchNet with an imbalance ratio of 100 from 28.24 to 31.30, a relative improvement of 10.83$\%$ compared to vanilla methods.

---

> > ### Author Response · Authors · 2025-11-25
> > **Response to Reviewer S1Zm**
> >
> > **6. Results of multi-attribute imbalance [W6, Q3]**
> >
> > Good questions. Following the reviewer's advice, we conducted a new experiment on datasets exhibiting multi-attribute imbalance (topic × length). The table below presents the average test accuracy  across six selection methods on AgNews and Yahoo datasets under two types of imbalance. The Table below shows that multi-attribute imbalance in annotated datasets significantly degrades the ICL performance and **our method can improve the ICL's performance under a multi-attribute imbalanced annotations**. We added the results in subsection 6.3 of revised version.
> >
> > |Length Imbalance Ratios|1||10||
> > |-|-|-|-|-|
> > |Topic Imbalance Ratios|1|10|1|10|
> > |Vanilla|66.57|62.29|63.25|62.68|
> > |**+Ours**|66.72|**65.94**|**66.34**|**65.22**|
> >
> > **7. Significant improvements with minimal computational overhead [W8]**
> >
> > Thank you for raising the concern. We would like to clarify that the improvements are significant and achieved with minimal computational overhead. **Table 1** reports average test accuracy and computational cost  with standard deviations (three runs) across six selection methods on four classification datasets with various imbalance ratios. The results show that our method improves ICL performance, and the improvement exceeds the reported standard deviations, thereby demonstrating statistical significance. For example, our method increases the average accuracy of Vanilla ICL from $46.47\pm0.88$ to $52.07\pm0.98$, yielding an absolute improvement of 5.60 percentage points. Furthermore, compared with existing calibration methods, our approach achieves an additional 2–5 accuracy improvement while requiring only 50% of their computational cost.
> >
> > [1] Bayesian optimization with inequality constraints. ICML  2014.
> >
> > [2] Searching for Optimal Solutions with LLMs via Bayesian Optimization. ICLR 2025.

---

### Author Response · Authors · 2025-11-25
**General Response  - part 1**

We thank all the reviewers for their time and valuable comments. We are glad that reviewers find this work focuses on an **useful** (Xduo) and **practical** (yh6B) problem.  We are encouraged that reviewers find that the method is **simple** (S1Zm, Xduo, Cf7b), **technically sound** (yh6B) and **computationally efficient** (yh6B, S1Zm). We are also  pleased that reviewers recognize that the experiments are **comprehensive** (SG7B, Cf7b) and **well-executed** (SG7B) with **good** (Xduo) and **consistent** (Cf7b, yh6B) improvements. Besides, reviewers recognize that this paper is **well-written**, **clearly structured** (Cf7b) and **straightforward to understand** (SG7B).

In the following responses, we have addressed the reviewers' comments and concerns point by point. The reviews allow us to strengthen our manuscript and the changes are summarized below:

* Added in-depth analysis of imbalanced annotations in **Line 183-191**. [SG7B, Xduo]
* Added experiments on RIF with different selected candidate sizes in **Line 304** and **Line 441-443**. [S1Zm, Cf7b, yh6B]
* Added experiments on datasets with extreme imbalance ratios in **Line 444-452**. [S1Zm, Cf7b]
* Added comparison with broader baselines in **Appendix G.2**. [yh6B]
* Added experiments on alternative importance-factor estimation strategies in **Line 453-460** , **Appendix G.3** and **G,4**. [S1Zm]
* Added experiments on multi-attribute imbalance in **Subsection 6.3**. [S1Zm]
* Added discussion of related studies in **Appendix C**. [Cf7b, yh6B]
* Add description of the construction of imbalanced datasets in **Line 366**. [yh6B]
* Fixed notations and typos in **Line 238**, **Line 1286**, **Line 1333**, **Line 1387**, **Line 1435** and added **Figure 8**. [Xduo, yh6B]

For clarity, we highlight the revised part of the manuscript in blue color.


**1. In-depth analysis of imbalanced annotations**

We provide a new analysis to explore how class imbalance affects ICL. Following the previous study [1], the effect of ICL can be decomposed into two components: Task Recognition (TR) and Task Learning (TL). TR measures the extent to which LLMs can recognize a task through ICL demonstrations, whereas TL reflects the ability to capture new input-label mappings unseen in pre-training. We conduct a new  experiment to show how class-imbalanced annotations affect the TR and TL components of ICL. The table below presents the average accuracy across the six selection methods on AgNews and Yahoo datasets with different imbalance ratios.

Our results show that TL consistently degrades as the imbalance ratio increases in ICL, while TR remains unchanged. TL relies on learning input–label mappings from the demonstrations; With the increase of imbalance rate, tail classes provide too few effective demonstrations to support such learning. In contrast, TR is insensitive to class imbalance since it mainly depends on recognizing the task format. This demonstrates that **Imbalanced annotations hurt the ICL performance by degrading the Task Learning ability.** We added the analysis of ICL regime in Line 183-191 of the revised version.

|Imbalanced Ratios|1|10|50|100|
|-|-|-|-|-|
|Task Recognition|34.60|34.98|34.68|33.88|
|Task Learning|64.40|53.96|45.12|40.98|

---

> ### Author Response · Authors · 2025-11-25
> **General Response - part 2**
>
> **2. Analysis of selected candidate sizes $K'$**
>
> In the main experiments presented in the paper, we set a large value of $K'$ to ensure that all classes (especially tailed classes) are included in the subset of candidates selected by existing methods. In the ablation study, we have conducted experiments with different values of $K'=\lambda\times K\times\phi$, where $K$ denotes the demonstration number, $\phi$ denotes the imbalance ratio, and $\lambda$ is a control factor (e.g. $\lambda=0.5, 1, 2$). **Figure 2 (d)** shows that the performance of our method is insensitive to the choice of $K'$ when $\lambda$ exceeds 1. Besides, we conduct a new experiment to report the computational cost (hours) of existing selection methods and our method on AgNews and Yahoo datasets using OPT-6.7B on a single NVIDIA L40 GPU. The table below shows that **our method does not incur additional computational cost** when $\lambda<1$,  because it can directly reuse the scores computed by existing selection methods. Therefore, we simply set the control factor $\lambda=1$ by default in our experiments.
> |Imbalanced Ratios|1|10|50|100|
> |-|-|-|-|-|
> |Vanilla|0.19|0.30|1.46|3.01|
> |$\lambda=0.5$|0.10|0.15|0.74|1.50|
> |$\lambda=1$|0.20|0.31|1.48|3.05|
> |$\lambda=2$|0.40|0.64|3.06|6.07|
>
>
>
> **3. Results of extreme imbalanced ratios**
>
> To demonstrate that our method remains effective even when tail classes contain very few examples, we conduct a new experiment on the AgNews and Yahoo datasets with extreme imbalance ratios (e.g., 500, 1000). We first employ an LLM-based augmentation method [2] to generate five times more examples that are semantically similar but phrased differently, based on the limited tail-class examples. presents the average accuracy across six selection methods on AgNews and Yahoo datasets with extreme imbalanced ratios. The results show that **our method works well in the cases of extreme imbalanced ratios** (e.g. 1000). Notably, the improvements are more significant in cases with larger imbalanced ratios. For example, when the imbalanced ratio increases from 100 to 1000, the improvement of our method increases from 8.21 to 10.58. We added the case of extreme imbalanced ratios in Line 444-452 of the revised version.
>
> |Imbalanced Ratios|100|500|1000|
> |-|-|-|-|
> |Baselines|53.77|46.80|44.52|
> |**+Ours**|**61.98**|**57.04**|**55.10**|
> |Improvement|+8.21|+10.24|+10.58|
>
> [1] What in-context learning “learns” in-context: Disentangling task recognition and task learning.
>
> [2] Empowering Large Language Models for Textual Data Augmentation. ACL 2024.

---

### Meta-Review · Area_Chair_BGpG · 2026-01-04

**Summary:**

In general, reviewers broadly agreed that the paper addresses a well-known problem that is intuitive: class imbalance in annotated data for ICL and that the authors have undertaken rigorous experiments for this problem statement. Moreover, most reviewers agree that both the problem formulation and the proposed solution offer limited novelty (Reviewers SG7B, Cf7b, and Xduo). In particular, I am inclined to agree with the reviewers' concern that the proposed RIF method is closely related to standard class reweighting and importance sampling techniques, and found the authors' response unconvincing in this regard. Reviewers also found the empirical gains to be modest and sometimes within noise, raising questions about the downstream practical impact of the proposed work. This could potentially be alleviated by undertaking statistical testing (i.e. hypothesis testing) as suggested by Reviewer S1Zm. Additionally, the paper’s positioning overstates its novelty by claiming to be the first to study imbalance in ICL, despite multiple relevant prior works that are not adequately discussed or contrasted (Reviewers Cf7b and yh6B). More specifically, while the authors mentioned that they incorporated this discussion on relevant work in Appendix C, a more thorough evaluation will be needed by reviewers to assess if this addition was indeed adequate or if the framing of the paper needs to further be improved. However, while the experimental rigor of the work is a definite plus, I believe making these changes and resubmitting to a new venue for consideration will be ideal. This revision should also account for the issues regarding presentation and clarity that several reviewers noticed.

**Reviewer Concerns:**

I believe the additional experiments conducted by the authors add to the experimental strengths of this work. However, the core concerns raised by multiple reviewers remain largely outstanding. In particular, the primary issue, i.e., limited novelty of the problem formulation as well as that of the proposed RIF method as an unsuprising balancing approach given the issue of class imbalance, was not convincingly resolved (Reviewers SG7B, Cf7b, and Xduo). The rebuttal also did not sufficiently justify the paper’s original framing as a first or systematic study of imbalance in ICL, despite the existence of several closely related prior works (Reviewers Cf7b and yh6B). Moreover, given the limited gains, the authors would benefit from including additional statistical analysis/testing of their results, such as via hypothesis testing (as suggested by Reviewer S1Zm). Overall, while clarifications were provided by authors, these mostly added to the empirical rigor of the work and in my view, the rebuttal did not materially change the assessment of the paper’s contribution or impact.

**Reviewer Scores:**

Reviewer S1Zm: I do not think the reviewer would have increased the score despite the additional experimental analysis as some of their concerns were not addressed (i.e. statistical testing given the limited gains).

Reviewer SG7B: The reviewer's concerns about the scope of the work were not addressed and as mentioned in their response to the authors, they decided to maintain their original score.

Reviewer Cf7b, Xduo, and yh6B: These reviewers had similar primary concerns about limited novelty of the problem statement and positioning within relevant literature. As these concerns were not sufficiently addressed in the rebuttal, I do not think the reviewers would have increased their scores.

---

### Decision · Program_Chairs · 2026-01-26

Reject